

Atmospheric
Measurement
Techniques

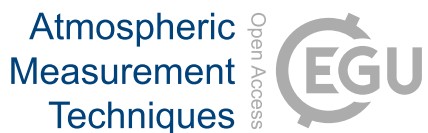

# Airborne measurements of CO$_2$ column concentrations made with a pulsed IPDA lidar using a multiple-wavelength-locked laser and HgCdTe APD detector

James B. Abshire[1], Anand Ramanathan[1,2], Haris Riris[1], Graham R. Allan[1,3], Xiaoli Sun[1], William E. Hasselbrack[1,3], Jianping Mao[1,2], Stewart Wu[4], Jeffrey Chen[4], Kenji Numata[4], Stephan R. Kawa[1], Mei Ying Melissa Yang[5], and Joshua DiGangi[5]

[1]Sciences and Exploration Directorate, NASA Goddard Space Flight Center, Greenbelt, MD 20771, USA
[2]Earth System Science Interdisciplinary Center (ESSIC), University of Maryland, College Park, MD 20740, USA
[3]Sigma Space Corporation, Lanham, MD 20706, USA
[4]Appl. Engineering and Technology Directorate, NASA Goddard Space Flight Center, Greenbelt, MD 20771, USA
[5]NASA Langley Research Center, Hampton, VA 23681, USA

**Correspondence:** James B. Abshire (james.b.abshire@nasa.gov)

**Abstract.** Here we report on measurements made with an improved CO$_2$ Sounder lidar during the ASCENDS 2014 and 2016 airborne campaigns. The changes made to the 2011 version of the lidar included incorporating a rapidly wavelength-tunable, step-locked seed laser in the transmitter, using a much more sensitive HgCdTe APD detector and using an analog digitizer with faster readout time in the receiver. We also improved the lidar's calibration approach and the XCO$_2$ retrieval algorithm. The 2014 and 2016 flights were made over several types of topographic surfaces from 3 to 12 km aircraft altitudes in the continental US. The results are compared to the XCO$_2$ values computed from an airborne in situ sensor during spiral-down maneuvers. The 2014 results show significantly better performance and include measurement of horizontal gradients in XCO$_2$ made over the Midwestern US that agree with chemistry transport models. The results from the 2016 airborne lidar retrievals show precisions of $\sim 0.7$ parts per million (ppm) with 1 s averaging over desert surfaces, which is a $\sim \times 8$ improvement compared to similar measurements made in 2011. Measurements in 2016 were also made over fresh snow surfaces that have lower surface reflectance at the laser wavelengths. The results from both campaigns showed that the mean values of XCO$_2$ retrieved from the lidar consistently agreed with those based on the in situ sensor to within 1 ppm. The improved precision and accuracy demonstrated in the 2014 and 2016 flights should benefit future airborne science campaigns and advance the technique's readiness for a space-based instrument.

## 1 Introduction

Accurate atmospheric CO$_2$ measurements with full global coverage are critically needed to better understand Earth's carbon cycle (Schimel et al., 2016). In order to allow atmospheric inversions to reduce uncertainties about carbon sources and sinks, studies show that space-based atmospheric column CO$_2$ mixing ratio (XCO$_2$) measurements need to have sub-ppm precision and biases on regional scales, with areas from 100 deg$^2$ (Tans et al., 1990; Fan et al., 1998; ESA A-SCOPE Report, 2008) to 1–25 deg$^2$ (NASA AS-CENDS Report, 2008). Several groups have analyzed space missions using passive spectrometers (Kuang et al., 2002; O'Brien et al., 2002; Dufour et al., 2003; Kuze et al., 2009), and the GOSAT (Yoshida et al., 2011) and OCO-2 missions (Crisp et al., 2017) are now making global XCO$_2$ measurements from space using optical spectrometers that view sunlit Earth.

However, there are limitations to XCO$_2$ measurements made using passive spectrometers. One inherent error source is optical scattering from aerosols and thin clouds in the illumination or observation paths (Mao and Kawa, 2004; Aben et al., 2007). Even small amounts of optical scattering in either path can modify the optical path length and thus the total CO$_2$ absorption measured and can cause large retrieval errors (Aben et al., 2007). For GOSAT, standard deviations (SDs) of 1.7 ppm were found vs. TCCON measurements with 0.5–0.8 ppm of that error irreducible by averaging, implying a bias of that order (Kulawik et al., 2016). For OCO-2 typical land measurements are found to have a precision and accuracy of approximately 0.75 and 0.65 ppm, respectively, based on the small region consistency assumption, which may well underestimate the bias between regions (Worden et al., 2017). A substantial portion of this error is likely related to interferences such as aerosols or surface albedo. Realistic simulations of the ACOS XCO$_2$ retrieval algorithm, used for both GOSAT and OCO-2, found errors of about 1 ppm in retrieved XCO$_2$, but, again, these are found to represent a lower limit on the errors present in retrievals using actual GOSAT observations (O'Dell et al., 2012). With the additional restriction from minimum required solar angles useful XCO$_2$ measurements from space with passive spectrometers have been restricted to daytime cloud-free scenes within the lower and midlatitudes.

To overcome these limitations, the US National Research Council's 2007 Decadal Survey for Earth Science recommended a space-based CO$_2$ measuring mission called ASCENDS (US National Research Council, 2007) that uses the laser absorption spectroscopy approach. The European Space Agency (ESA) also previously carried out mission definition studies for a similar space mission called A-SCOPE (ESA A-SCOPE Report, 2008; Durand et al., 2009) and has supported lidar sensitivity and spectroscopic analyses for it (Ehret et al., 2008; Caron et al., 2009). The ASCENDS mission's goals are to quantify global spatial distribution of atmospheric column XCO$_2$ with < 1 ppm accuracy and to quantify the global spatial distribution of terrestrial and oceanic sources and sinks of CO$_2$ with monthly time resolution. The lidar approach directly measures range to the surface along with CO$_2$ absorption and can provide XCO$_2$ measurements through thin clouds and aerosols. The measurement is independent of sun angle and scattered light has little impact. It provides continuous coverage of land and ocean daytime and nighttime. The ASCENDS mission organizers held an initial workshop in 2008 to define the science and measurement needs and to develop plans for future work (NASA ASCENDS Report, 2008). In 2015 the study team summarized their result in a white paper (NASA ASCENDS White Paper, 2018 TS1) along with plans for future work.

The integrated path differential absorption (IPDA) lidar technique is based on laser absorption spectroscopy and has been widely used for open-path measurements of atmospheric gases (Measures, 1992; Weitkamp, 2005). Several groups have developed IPDA lidar for airborne measurements of XCO$_2$ using different types of laser sources, detection and analysis techniques. Examples of lidar that have targeted measuring a single CO$_2$ line in the 1570 nm band include two airborne lidar that use intensity-modulated continuous wave (CW) lasers and direct detection receivers (Dobler et al., 2013; Lin et al., 2015; Obland et al., 2015). Another is a pulsed airborne IPDA lidar (Amediek et al., 2017) that simultaneously measures the CO$_2$ absorption near 1572 nm and CH$_4$ absorption near 1646 nm using a direct detection receiver. Examples of lidar that have targeted the 2051 nm CO$_2$ line include a two-wavelength laser absorption spectrometer using CW lasers and heterodyne detection (Spiers et al., 2011, 2016; Menzies et al., 2014) and a pulsed lidar that measures CO$_2$ absorption with two or three wavelengths (Refaat et al., 2015; Yu et al., 2017). Several studies have also investigated the benefits and feasibility of developing a lidar to measure XCO$_2$ from orbit; discussed options for orbits, the laser transmitter, the needed laser power and receiver approaches; and have estimated measurement performance (NASA ASCENDS workshop, 2008; Kawa et al., 2010; Singh et al., 2017; Han et al., 2017; NASA ASCENDS White paper, 2018 TS2).

## 2   The airborne CO$_2$ Sounder lidar

The airborne CO$_2$ Sounder lidar (Riris et al., 2007; Abshire et al., 2010a, b; Amediek et al., 2012) was developed to demonstrate a pulsed multi-wavelength IPDA approach as a candidate for the ASCENDS mission. Its configuration and performance in the 2011 ASCENDS campaign are described in Abshire et al. (2013a, b). The pulsed transmitter approach allows simultaneous measurement of the absorption of a single CO$_2$ line in the 1570 nm band and the atmospheric backscatter profile and scattering surface height(s) in the same path. The laser transmitter uses a tunable diode laser followed by a modulator to produce pulses and a series of laser amplifiers. The direct detection receivers measure the time-resolved backscattered laser energy from the atmosphere and the surface. The column average CO$_2$ concentration is estimated from the pulse energies of the surface returns via a retrieval algorithm. It uses the lidar sampled transmission wavelengths, the aircraft altitude, the measured range to the scattering surface, line spectroscopic data and a layered model for atmospheric state to calculate the best-fit XCO$_2$ value to the lidar signals.

The CO$_2$ Sounder measurement samples a single CO$_2$ line in the 1570 nm band (Mao and Kawa, 2004). This vibration–rotation band of CO$_2$ has an appropriate range of absorption that provides good sensitivity to the surface echo signal and to variation in CO$_2$ in the lower troposphere. The band has minimal interference from other atmospheric species like H$_2$O and has several temperature-insensitive lines. Although using other lines in this band is also possible, the R16 line at 1572.335 nm has been analyzed and was found attractive for

CO$_2$ measurements (Mao et al., 2007). It has low temperature sensitivity, particularly to changes in the lower atmosphere.

The CO$_2$ Sounder approach samples the CO$_2$ line shape at multiple wavelengths. This provides several benefits including extracting line shape and some information on the vertical CO$_2$ distribution in the retrievals. It also allows solving for useful spectroscopic information, such as line center wavelengths, line widths and errors in the fits (Ramanathan et al., 2013). This approach also provides information that allows solving for several different measurement environmental variables and instrument parameters, such as Doppler-shift and wavelength offsets, baseline tilts and wavelength-dependent instrument transmission. Our work has found that this information is essential to minimize biases in the XCO$_2$ retrievals. For airborne and space measurements, performing retrievals in the presence of Doppler shifts expands the instrument capability to allow continuous measurement at off-nadir pointing angles during maneuvers or when pointing at ground targets.

There were several factors that led to the choice of the pulsed approach, laser pulse rate and pulse width. Using lower pulse energies at a higher pulse rate enables the use of fiber-based technology throughout the laser transmitter. At higher laser pulse rates, there are also a larger number of receiver measurements in a given time, which allows using more averaging to reduce speckle noise. Using pulsed lasers also allows post-detection signal processing to isolate the laser echo signals from the primary scattering surface and to reject backscatter from the atmosphere that arrives earlier. Hence it allows isolating the full column measurement from potential bias errors caused by atmospheric scattering (Mao and Kawa, 2004; Aben et al., 2007). It also allows useful XCO$_2$ measurements to the tops of clouds (Ramanathan et al., 2015; Mao et al., 2018). Isolating the surface reflected pulse from the atmosphere backscatter profile also substantially improves the receiver's signal-to-noise ratio (SNR) by limiting the amount of noise from the detector and solar background.

A previous version of the lidar was used in the 2011 AS-CENDS airborne campaign (Abshire, 2013b). This previous version had a similar basic design to the one reported here. However, its seed laser source was not locked, but rather the center of its pulsed wavelength scan was periodically calibrated by using a reference laser whose frequency was monitored with a wavemeter. It also used a much less sensitive photomultiplier (PMT) detector followed by a discriminator and a photon counter in its receiver. After the 2011 campaign a detailed analysis was made on four flights that flew over a variety of surface and cloud conditions near the US. These included over a stratus cloud deck over the Pacific Ocean, to a dry lake bed surrounded by mountains in Nevada, to a desert area with a coal-fired power plant, from the Rocky Mountains to Iowa, and over cloud land with both cumulus and cirrus clouds. Most flights had five to six altitude steps to > 12 km. Analyses of the 2011 measurements showed the re-

trievals of lidar range, CO$_2$ column absorption and CO$_2$ mixing ratio worked well when measuring over topography with rapidly changing height and reflectivity, through thin clouds, between cumulus clouds and to stratus cloud tops (Mao et al., 2018).

The measurement precision of the 2011 version of the lidar was limited by the linear dynamic range of the PMT detector and by the signal photon count of the laser wavelengths on the CO$_2$ absorption line. For 10 s averaging, the scatter in the 2011 retrievals was typically 2–3 ppm. The analysis showed the differences between the mean lidar-retrieved values, based on the DC-8 measured atmosphere, and the in situ measured CO$_2$ column concentrations to be < 1.4 ppm for all four flights at altitudes > 6 km.

## 3   CO$_2$ Sounder lidar used in 2014 and 2016 campaigns

Photographs of the lidar are shown in Fig. 1. For these campaigns the lidar's transmitter–telescope unit was mounted above the NASA DC-8's (NASA DC-8 Fact Sheet, 2017) aft-most nadir window (Port 9). The window assembly used separate wedged and antireflection-coated optical windows for the transmitter and receiver. The laser transmitted pulses at a 10 kHz rate while the wavelengths of the laser pulses are sequentially stepped across the 1572.33 nm (6360 cm$^{-1}$) CO$_2$ absorption line. Although the number of laser wavelength steps is programmable, all airborne campaigns to date have used either 30 or 15 steps. The receiver telescope collects the backscatter and focuses it onto the receiver detector. The detector's analog output is amplified and digitized, and the data are synchronously averaged and recorded.

After the 2011 flight campaign, our team made a set of improvements to that version of the CO$_2$ Sounder lidar (Abshire et al., 2013b). The parameters for the 2011, 2014 and the 2016 versions are summarized in Table 1. For the 2014 flights, we replaced the previous wavelength-swept seed laser source with a rapidly tunable step-locked seed laser (Numata et al., 2012). Figure 2 shows a block diagram of the lidar configuration used in the 2014 and 2016 airborne campaigns. For these campaigns the wavelength settings of the seed laser were locked and better optimized for measuring the CO$_2$ absorption lineshape. In the lidar receiver, we increased the receiver's optical transmission and replaced the PMT-based photon-counting receiver with a much more sensitive 16-element HgCdTe avalanche photodiode (APD) detector whose analog output was recorded by an analog digitizer. This change also increased the lidar receiver's linear dynamic range and readout rate from 1 to 10 Hz. In 2016 we also increased the laser's divergence and the receiver field of view to reduce speckle noise. Finally we improved the retrieval algorithms and models that solve for range for parameters that can cause offsets in the measurements and in XCO$_2$ retrievals. Together all these changes considerably improved the lidar's measurement precision, stability and dynamic range and reduced measurement bias.

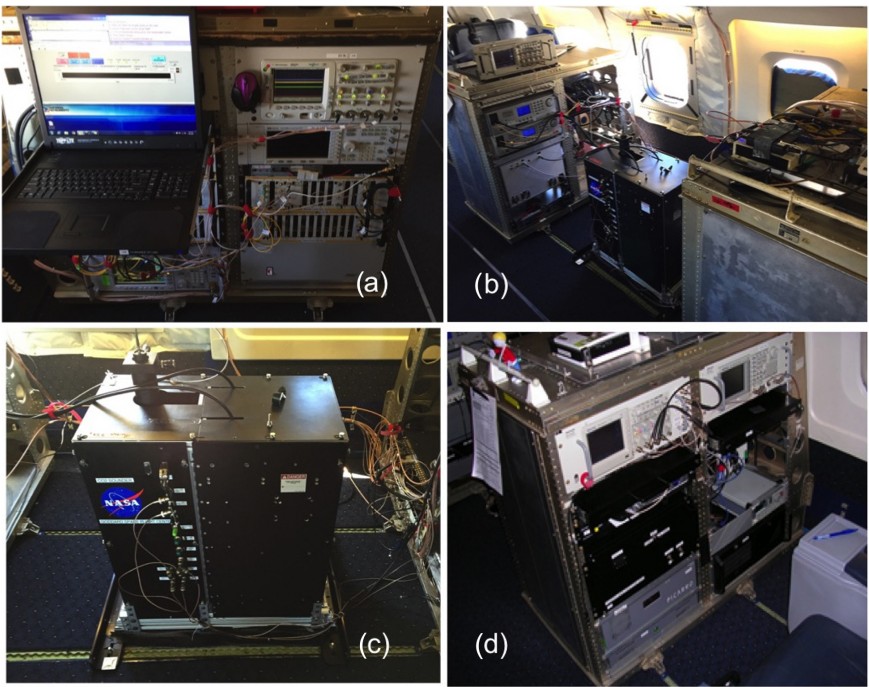

**Figure 1.** CO₂ Sounder instrument photographs. (**a**) The aircraft rack with the new seed laser subsystem. (**b**) The aircraft racks containing the laser power amplifiers and the lidar's detector subsystem. (**c**) The lidar transmitter and receiver telescope assembly, which is positioned over the nadir window assembly in the aircraft fuselage. The optical pulses from the fiber amplifiers and the received optical signals are coupled via fiber optics. (**d**) The instrument operator's console, with the control computer screens folded away.

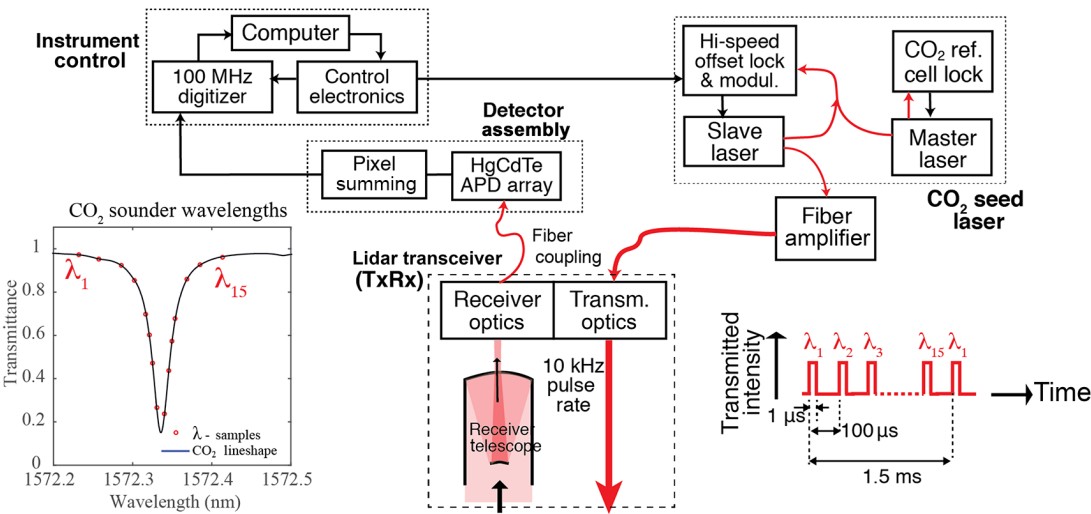

**Figure 2.** Instrument block diagram for the 2014 and 2016 versions of the CO₂ Sounder lidar described here. The inset on the right shows the transmitted pulse train sequence that is repetitively stepped in wavelength across the CO₂ line.

Figure 3 shows more detail on the design of the CO₂ seed laser subsystem used in the 2014 and 2016 campaigns (Numata et al., 2012). The master laser (a single-frequency DFB laser diode) was continuously locked to the peak of the 1572.335 nm line of CO₂ in the Herriott absorption cell via the Pound–Drever–Hall technique (Numata et al.,

2011). The cell pressure was 40 mb and optical path length was 18 m. A single-frequency slave laser (a DS-DBR laser diode) was dynamically offset-frequency-locked to the master laser using a rapidly tunable, step-locked, phase-locked loop technique (Numata et al., 2012). The offset frequencies were supplied by the FPGA. The resulting frequency-

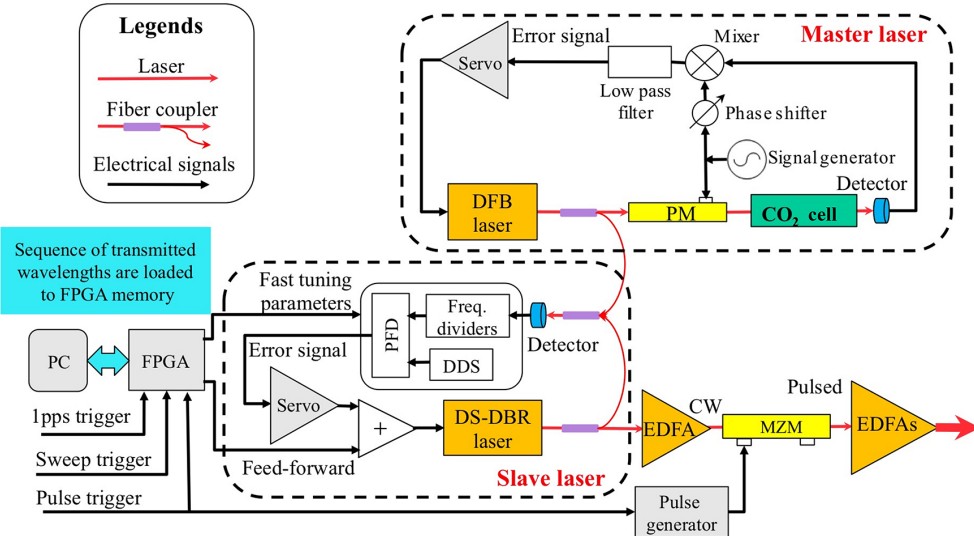

**Figure 3.** Block diagram of the CO$_2$ seed laser subsystem that is used to produce the wavelength-stepped pulse train transmitted by the lidar. The wavelength of the master laser (a DFB laser diode) is frequency locked to the center of the CO$_2$ absorption line. The slave laser is offset-frequency-locked to the master via an optical-phase-locked loop. The frequency offset is changed during the 100 us between laser pulses based on the wavelength settings stored in a table in the seed laser's FPGA. The slave laser's output is carved into 1-us wide pulses by the modulator (MZM) and is used as the input for the transmitter's fiber amplifiers (EDFAs) that produce the laser pulse train that is transmitted. Here PM denotes phase modulator; PFD, phase frequency detector; FPGA, field programmable gate array; DDS, direct digital synthesizer; MZM, Mach–Zehnder modulator; and EDFA, erbium doped fiber amplifier.

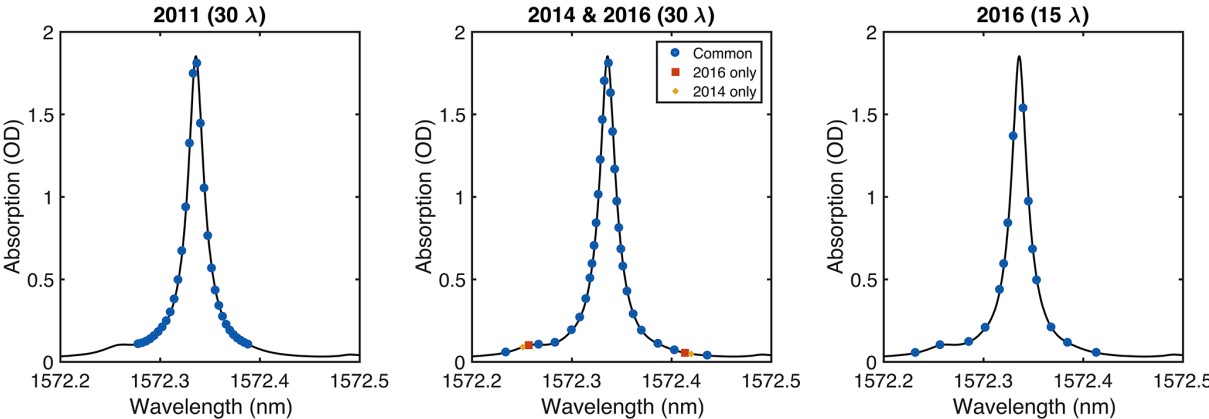

**Figure 4.** Plots of the CO$_2$ line sampling laser wavelengths (blue dots) used for the airborne campaigns in 2011, 2014 and 2016. The 2016 campaign used both 30 and 15 laser sampling wavelengths. The CO$_2$ absorption line shapes (black lines) are shown for a two-way path for airborne lidar measurement conditions from a flight altitude of 13 km to a surface elevation of 220 m. These conditions occurred during the 3 September 2014 flight over Iowa.

stepped CW output from the slave laser was modulated into a 10 kHz pulse train by an electro-optic modulator, amplified by a commercial erbium doped fiber amplifier (EDFA) and collimated and transmitted.

The receiver's Cassegrain telescope viewed nadir through the receiver window and collected the laser backscatter. An antireflection-coated multi-mode optical fiber was used to couple the signal from the telescope focal plane to the receiver optics. After passing through an optical bandpass fil-

ter, the signal was focused onto a $3 \times 3$ pixel area of the $4 \times 4$ pixel HgCdTe APD detector. The electrical outputs from the $3 \times 3$ pixels were amplified, summed together, passed through a low-pass filter and digitized at a 100 MHz rate. The start time of the digitizer recording sweep was synchronized with the trigger for start of the laser wavelength sampling sequence. The receiver electronics averaged the signal for 32 wavelength sampling sequences (64 when using 15 samples), storing them in the memory, resetting itself and starting

**Table 1.** $CO_2$ Sounder lidar parameters for the 2011, 2014 and 2016 airborne campaigns.

| Parameter | 2011 flights | 2014 flights | 2016 flights |
|---|---|---|---|
| $CO_2$ line used | R16, 6359.96 cm$^{-1}$ | same | same |
| $CO_2$ line center wavelength | 1572.335 nm | same | same |
| Laser min wavelength | 1572.228 nm | 1572.235 nm | same |
| Laser max wavelength | 1572.39 nm | 1572.440 nm | same |
| Laser pulse rate | 10 kHz | same | same |
| No. of wavelength samples on line | 30 | same | 30 or 15 |
| Laser scan rate of $CO_2$ line | 300 Hz | same | 300 or 600 Hz |
| Seed laser wavelength adj. | linear sweep | step locked | same |
| Wavelength change/laser step | $\sim 3.8$ pm | varied via program | same |
| $CO_2$ reference cell conditions | 0.8 m path, $\sim 200$ Torr pressure | 18 m path, 40 mbar pressure | same |
| Laser peak power, pulse width | 25 watts, 1 µ sec | same | same |
| Primary laser pulse energy | 25 µJ | same | same |
| Optional laser pulse energy[a] | – | – | 50 µJ |
| Laser divergence angle | 100 µrad | 100 µrad | 430 µrad |
| Laser linewidth | $\sim 15$ MHz | $< 4$ MHz $= 0.032$ pm | same |
| Receiver telescope type | Cassegrain, f/10 | same | same |
| Telescope diameter | 20 cm | same | same |
| Receiver field-of-view diameter | 200 µrad | 200 µrad | 500 µrad |
| Receiver optical transmission | $\sim 50\%$ | 9.2 % | 60 % |
| Detector type | Hamamatsu H10330A-75 | DRS HgCdTe APD | same |
| Detector effective QE | 4 % | 70 % | 70 % |
| Detector gain | $\sim 10^5$ | 600 | 300 |
| Receiver signal processing approach | Photon counting and histogramming | Analog detection and averaging | same |
| Receiver time bin width or ADC sample time bin width | 8 ns | 10 ns | same |
| Receiver electronic bandwidth | 10 MHz | 7 MHz | same |
| Data recording rate | 1 Hz | 10 Hz | same |
| Data recording duty cycle | 80 % | 80 % | 90 % |

[a] Used two laser amplifiers.

recording again at the beginning of the next 100 ms. The laser trigger and the data acquisition were synchronized to timing markers from the GPS receiver and data were stored every 0.1 s. The computer also recorded other signals, including the GPS position and time. Due to the computer time needed to store data, not all received profiles could be recorded, and the duty cycles for the stored data were 80 and 90 % for the 2014 and 2016 campaigns, respectively. The DC-8 data system also recorded many other parameters, including aircraft position, altitude and pitch and roll angles, that were later used in data analysis and $XCO_2$ retrievals.

Figure 4 shows the wavelength sampling of the $CO_2$ line shape used in the 2011, 2014 and 2016 campaigns. It shows that wavelength samples in 2014 and 2016 were more widely distributed in wavelength and were also more uniformly distributed in optical depth. Both changes improved the retrieval results. In the 2016 flights we also made some measurements using the 15 wavelength samples shown in the figure. The receiver optics had some variability in spectral transmission that impacted the lidar measurements. Plots of the optical transmission vs. wavelength for the optical bandpass filters

used to reduce solar background in the lidar receiver are shown in Fig. 5. The insets show expanded views with the red dots indicating the lidar measurement wavelengths. Since the transmission is not uniform with wavelength for the $CO_2$ measurement region near the peak of filter's transmission, the bandpass filter slightly distorts the measured $CO_2$ lineshape. This distortion in transmission is solved for as part of the lidar retrieval algorithm.

Figure 6 shows the 16-element HgCdTe APD lidar detector (Sun et al., 2017a, b) used in the 2014 and 2016 flights. The detector and preamplifier chip were cooled to 77 K and were housed in a commercial integrated Dewar cooler assembly. A multimode fiber optical cable coupled the optical signal from the telescope to the detector assembly and the signal was focused onto the detector array through an optical window. For the 2014 campaign an extra fiber optical assembly was used between the telescope and the detector. This was later found to introduce excessive losses and so reduced the receiver's optical transmission for that campaign. This assembly was removed and the receiver's optical transmission was re-optimized for the 2016 campaign.

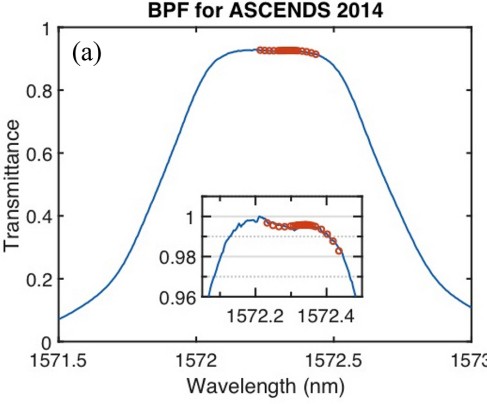

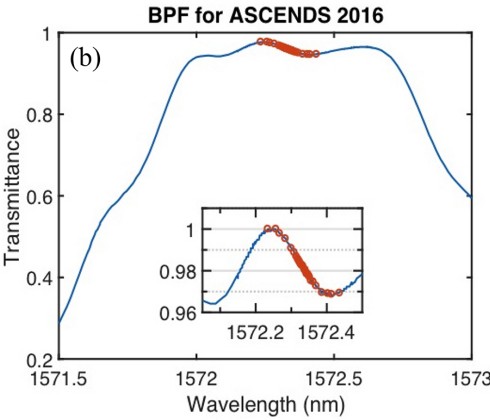

**Figure 5.** Plots of transmission vs. wavelength for the optical band-pass filters (BPF) used in the lidar receiver. **(a)** Filter used for the 2014 flights and **(b)** filter used for the 2016 flights. The insets show expanded views of the peak filter transmissions at the lidar measurement wavelengths (red dots). The 2016 filter was purchased in an attempt (which was unsuccessful) to flatten the response at the lidar measurement wavelengths. The lidar retrieval algorithm solves for the variability in instrument transmission at the measurement wavelengths introduced by the filters.

For unbiased XCO$_2$ measurements the lidar detector's output voltage must respond to optical power in a highly linear fashion. Figure 7 shows the results from evaluating the dynamic range and the linearity of the HgCdTe APD detector for the 2014 flights, before the optical illumination of the pixels was optimized. The detector response was linear until 500 detected photons and the nonlinearity slowly grows to 1 % at 2000 detected photons. This correction factor was also solved for as part of the 2014 retrieval algorithm. For the 2016 flights, the receiver's optical focus was better optimized so that the detector pixels were illuminated much more uniformly. Also during the 2016 flights the laser transmitter energy was reduced for the lower altitude measurements. The changes greatly reduced the peak powers on some pixels, that a detector nonlinearity correction was not required for the 2016 campaign.

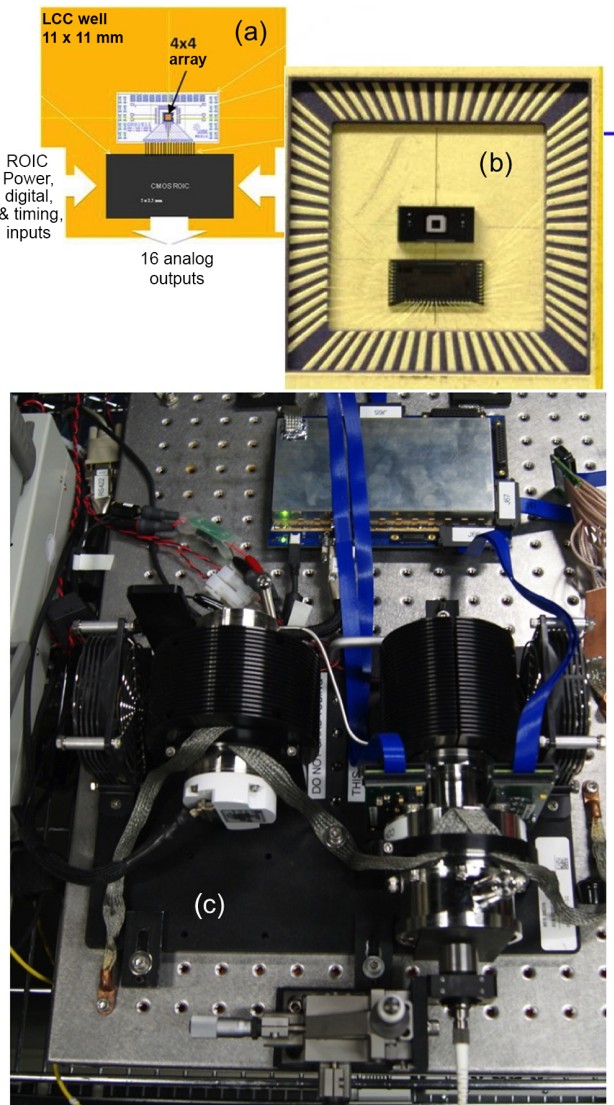

**Figure 6.** The HgCdTe APD detector used in the lidar receiver. **(a)** A diagram showing the locations of the 4×4 APD detector array and the CMOS preamplifier and readout integrated circuit (ROIC). **(b)** Photograph of the same elements mounted on the detector's lead-less chip carrier (LCC). **(c)** Top view of the detector's cryo-cooler assembly used to keep the LCC at ∼ 80 K. The cooler's compressor is on the left, the cooled section is on the right, and the fiber optic cable used to couple the optical signal from the telescope through the cryo-cooler's optical window assembly onto the 4 × 4 detector array is at the bottom. The conditioning and control electronic box is at the top of the photograph.

## 4  Data processing and XCO$_2$ retrievals

The retrieval algorithm approach is shown in Fig. 8. First, the receiver backscatter at 10 Hz is further averaged over 1 s and then searched for pulse echoes with significant energy, such as those reflected from cloud tops or from the ground. The averaged pulse echo energies at each wavelength

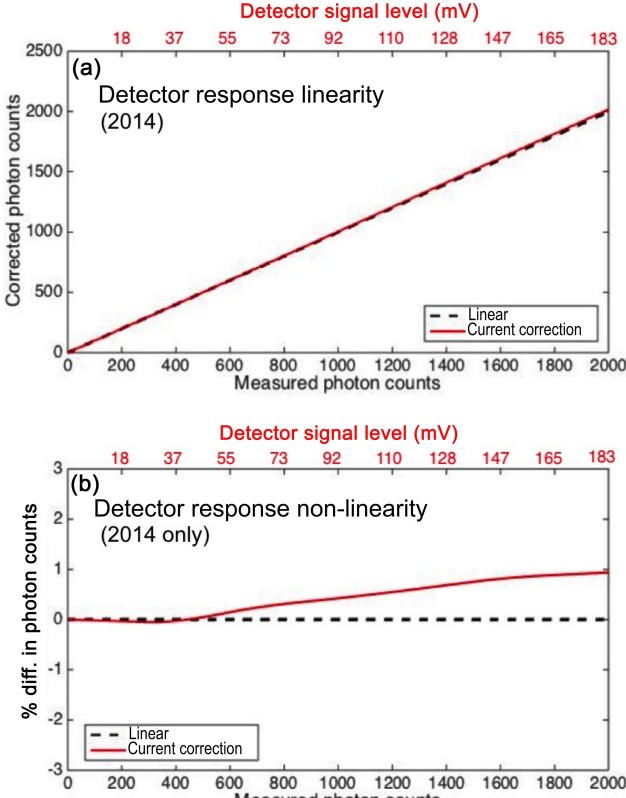

**Figure 7.** The results from calibrating the linearity of the lidar's HgCdTe detector for the 2014 lidar configuration. **(a)** Measurement of the detector output vs. optical input. **(b)** Deviation from linear response in the detector response, showing a deviation starting at 600 photons with a maximum deviation of 1 %. For the 2016 flights, the receiver optics were improved so that the optical signal was more uniformly distributed across the detector array elements, and an additional electronic preamplifier stage was used. Together these reduced the nonlinearity effect so that its effect was negligible for the 2016 flights.

are then corrected for variation in transmission of the receiver's optical bandpass filter and for any detector nonlinearity. The calibrated pulse echoes are then normalized by the transmitted laser energy and divided by the square of the range to yield the product of transmission and surface reflectance at all 30 wavelengths. This yields a first estimate of the lidar-sampled $CO_2$ transmission line shape. The 1 s averaged transmittances across the $CO_2$ absorption line are then converted into optical depth (OD), which is linearly proportional to number density of $CO_2$.

Flight calibrations are constructed from a segment during the engineering flight that had known atmospheric conditions and a vertical profile of $CO_2$ mixing ratio measured by the in situ sensor during the flight's spiral-down maneuver. Radiative transfer calculations are used to predict the $CO_2$ transmission line shapes at different altitudes based on the in situ $CO_2$ measurements. This allows solving for and apply-

ing any further corrections needed to compensate for instrument changes seen in flight, such as for detector nonlinearities and for any changes in the wavelength dependence of the receiver optics. These final calibrations are then applied to all retrievals for the science flights.

### Line shape and fit

The retrievals utilize a $CO_2$ absorption line shape based on atmospheric state information (pressure, temperature and water vapor profiles) from the near-real-time forward processing data of the Goddard Earth Observing System Model, Version 5 (GEOS-5) (Rieneker et al., 2011). Data on the full model grid (0.25° latitude × 0.3125° longitude ×72 vertical layers, every 3 h) are interpolated to flight ground track position and time. The aircraft altitude, measurement path angle and altitudes of the significant scattering surfaces are determined using the aircraft GPS altitude, pitch and roll angles and the lidar-measured range. For the $CO_2$ line shape calculation, the algorithm used the spectroscopy database HITRAN 2008 (Rothman et al., 2009; Lamouroux et al., 2010) and the Line-By-Line Radiative Transfer Model (LBLRTM; Clough et al., 1992, 1995) V12.1 to calculate $CO_2$ optical depth and create look-up tables (LUTs). These are initially computed for a vertically uniform 400 ppm mixing ratio.

The algorithm then retrieves the best-fit XCO₂ by comparing the line shapes calculated based on the vertically uniform mixing ratio to the lidar-measured line shape samples. The algorithm performs the line shape fitting in optical transmission using an unconstrained minimum variance fit. At each measurement wavelength, the fitting residual is weighted by the square of estimated SNR at that wavelength based on the average received signal and the instrument model. The retrieval algorithm then solves for Doppler shift, baseline offset, slope, surface reflectance and XCO₂ simultaneously by using a best fit of the lineshape function to the data. The Level 2a products are created at this step are shown in Fig. 8. An example of the transmission line shape and the results of the line fitting process are shown in Fig. 9.

### Weak water vapor lines

There is also a weak (OD ∼ 0.01 to 0.05) isotopic water vapor (HDO) absorption line on the short wavelength shoulder of the 1572.335 nm $CO_2$ line currently measured by the lidar, as well as one about 4 times weaker near 1572.389 nm. When measuring this $CO_2$ line, the HDO absorption spectrum can bias the retrieved XCO₂ value by up to 2 ppm if not taken into account. Our laser transmitter wavelength assigned one or two wavelengths on the short wavelength HDO line to allow solving for column water vapor concentration (XHDO). The XHDO retrievals are used iteratively to reduce the uncertainty of the water vapor content in the forward calculations and then to improve the XCO₂ retrievals.

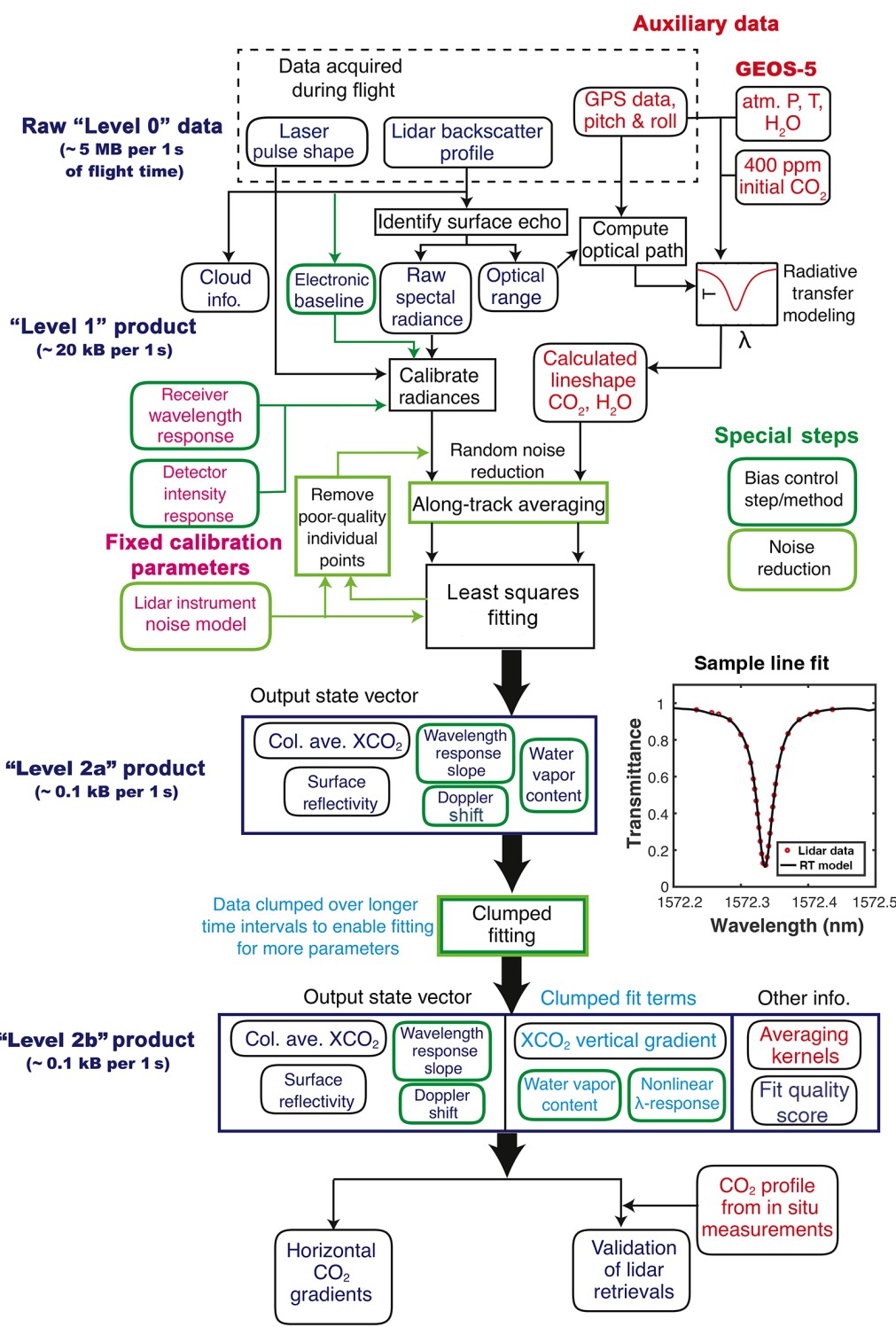

**Figure 8.** Processing diagram for retrieval algorithm used to estimate XCO$_2$, range and other parameters from the lidar measurements as well as from other information from the aircraft. The results shown in this paper are labeled as Level 2a and Level 2b products in the algorithm.

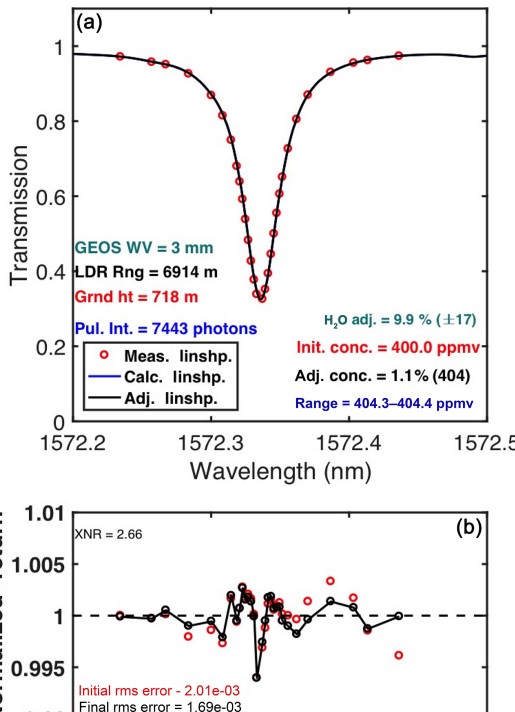

**Figure 9.** (a) Example of the CO$_2$ transmission line shape measured by the lidar from an altitude of 7.6 km. The line shape samples from the lidar are the red dots. The line shape computed from the retrieval is shown as the black line. (b) The ratio of the retrieved line shape and lidar retrievals with the red dots being the initial trial XCO$_2$ value of 400 ppm and the black dots and line being the final best-fit retrieved XCO$_2$ value of 404 ppm.

However, the interfering HDO lines are weak and do not have other properties, such as temperature insensitivity, favorable to accurately retrieving column HDO. We estimate the precision of our retrievals of the HDO column (assuming only random noise), based on our posterior uncertainties to be around 1 %. Hence the accuracy of the HDO column is likely worse, especially considering the variability in the water vapor profile. To date the HDO retrievals have been useful for pointing toward errors in the model water vapor columns. Further study may yield opportunities for using the HDO retrievals in comparison to meteorological models or to study water cycle processes.

**Clumped fitting**

Retrievals of XCO$_2$ with high ($\sim$ parts per thousand) precision require line fits with very small residual errors. This requires the retrieval algorithm to solve for several potential sources of systematic error. Some of these, like the receiver wavelength response and detector intensity response, can be carefully calibrated. Other sources, like changes in the water vapor column, the Doppler shift or slow instrument drifts in the receiver wavelength response are time-varying and so

cannot be addressed using calibrations. Still other sources of systematic error, such as the Doppler shift or changes in the receiver's wavelength response, have impacts on the line fits that are orthogonal to changes in XCO$_2$. This allows them to solved for in the line fitting process without impacting the XCO$_2$ retrieval. The remaining systematic errors have some overlap with the CO$_2$ line shape, and so fitting for them in each line fit can cause an increase in the XCO$_2$ uncertainty. One example of this is that the Level 2a fitting removes potential bias from the HDO line by fitting for the water vapor. If this is performed for each fit, this comes at the cost of a 30 % increase in the SD of XCO$_2$.

We addressed these systematic errors by using clumped fitting. Clumped fitting takes advantage of temporal correlations of some systematic errors and attempts to minimize their effect on the line fit without substantially impacting the XCO$_2$ posterior uncertainty or averaging kernel. Clumped fitting works similarly to the multi-pixel retrievals used by AIRS (Susskind et al., 1998 and 2003; Langmore et al., 2013) and post-retrieval processing used by TCCON (Wunch et al., 2011) and GOSAT/OCO-2 (O'Dell et al., 2012) to lower biases and reduce scatter. However, rather than assuming varying degrees of correlation between different soundings, our algorithm uses a single, averaged value for the entire clump. A typical case for the 2016 flights were XCO$_2$ retrievals to data with 30 wavelength samples averaged over 1 s. In this step, 20 s clumps of 1 s retrievals (20 line shapes) are simultaneously fit for the parameters of the above-mentioned systematic effects being held fixed for the entire clump, while the remaining parameters, including XCO$_2$, are allowed to vary on a 1 s scale. This gives a state-vector size (fitting parameters) of $4 \times 20$ individual terms + 3 clumped terms = 83 total terms. The measurement basis for the clumped fit is 30 wavelengths $\times$ 20 line-shapes = 600 samples. In contrast, the Level-2a fitting terms had 30 wavelength samples and 5 fitting terms for each of the 20 line shapes.

In implementing clumped fitting, we found it is important to exclude the Level 2a line fits that had high residual errors (for example, from very high signals from specular reflections from smooth water surfaces). Our analysis of clumped fitting showed it was able to remove small biases from the systematic effects mentioned above, with little change to the uncertainty in the retrieved XCO$_2$ or its averaging kernel. The averaged values of 1 s XCO$_2$ retrievals from Level 2a processing are then adjusted for these terms. After this processing, the retrievals, now called Level 2b products, had smaller errors than those from Level 2a.

## 5 Overview of airborne campaigns

Table 2 summarizes the flight locations, focus of measurements, flight altitudes and number of lidar measurements for the 2014 and 2016 flights reported here. All flights were based out of NASA Armstrong Flight Facility in Palmdale,

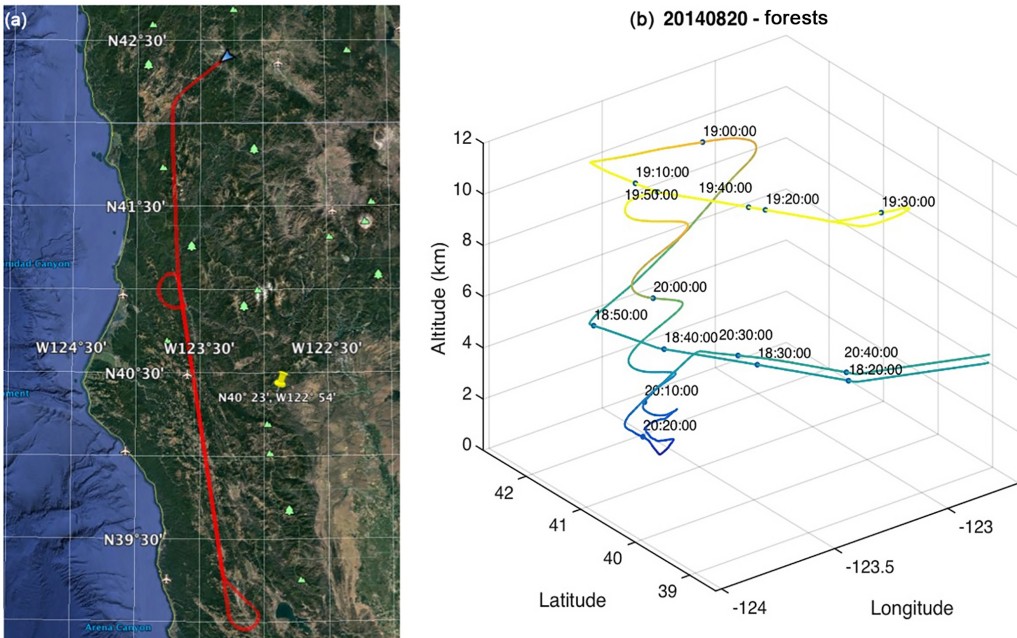

**Figure 10. (a)** Map of the 2014 SF1 flight track over the northern California coast on 20 August 2014. **(b)** Time-tagged location and altitude plot for the same flight.

**Table 2.** Summary of 2014 and 2016 campaign flights and regions studied.

| Flight designation | Date | Location (in US) | Focus of measurements | Aircraft altitude range (km) | Ave. time per measurement (s) | Number of lidar measurements |
|---|---|---|---|---|---|---|
| 2014 SF1 | 20 Aug 2014 | North CA coastal forests | Forests on low mountains | 2.89–11.19 | 10 | 712 |
| 2014 SF2 | 22 Aug 2014 | Near Edwards AFB, CA | Desert through haze | 3.50–11.25 | 10 | 446 |
| 2014 SF5 | 3 Sep 2014 | Eastern Iowa | XCO$_2$ over cropland | 2.62–11.16 | 10 | 1010 |
| 2014 SF3G1 | 25 Aug 2014 | Colorado to Iowa (outbound) | East–west XCO$_2$ gradients | 11.2 | 50 | 43 |
| 2014 SF3G2 | 25 Aug 2014 | Iowa to Colorado (return) | East–west XCO$_2$ gradients | 5.6, 6.3, 10.8 | 50 | 67 |
| 2016 desert | 10 Feb 2016 | Edwards AFB, CA | Desert | 3.59–12.60 | 1 | 1281 |
| 2016 snow | 11 Feb 2016 | Eastern Nevada | Recent cold snow | 6.68–9.52 | 1 | 5893 |

CA. As in previous ASCENDS campaigns, for each flight we compared lidar measurements of XCO$_2$ made during spiral-down maneuvers to the surface with those computed from the AVOCET in situ sensor (Choi et al., 2008; Vay et al., 2003). Lidar measurements were made over low mountains covered by tall trees, desert areas with atmospheric haze, areas with growing crops, a transition area between high plateau and cropland, fresh cold snow and clear sky over desert. Spiral-down maneuvers were made over most types of areas, allowing the lidar retrievals of XCO$_2$ to be compared to the column average from in situ sensors.

The retrieval results, described subsequently, show the lidar worked well during both campaigns, although the 2016 airborne results were best due to the higher receiver optical transmission and the reduced speckle noise. The retrievals from the 2016 measurements made over desert surfaces from a 10 km altitude with 1 s averaging time consistently had

a SD of ∼ 0.8 ppm, while those with 10 s averaging time had precision of 0.3 ppm. This is a 5-fold improvement in precision over measurements made in 2011 (Abshire et al., 2013b), where the agreement between the lidar and in situ values of XCO$_2$ were < 1.4 ppm. The higher precision in 2016 also allows a more careful comparison of differences in lidar-measured XCO$_2$ values to those computed from the column averaged in situ sensor. In most cases, the agreement of average XCO$_2$ computed from the lidar to that computed from the in situ sensor was better than 1 ppm.

## 6  2014 airborne campaign

### 2014 SF1

The focus of the 2014 Science Flight 1 (SF1) was to make measurements over a forested region with tall trees and targeted the northern California coast. The ground track for the

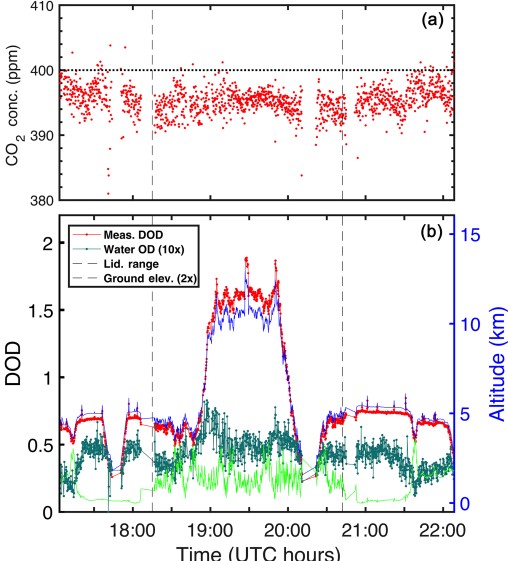

**Figure 11.** Lidar measurement and retrieval results from 2014 SF1 flight over the northern coast of CA on 20 August 2014. **(a)** The retrieved XCO$_2$ values from the lidar measurements, with each dot made using 10 s averaging time. **(b)** Time-resolved results showing the lidar-measured differential optical depth in red, the range to the surface in blue and the computed elevation of the scattering surface in dark green. The upward spikes in the DOD and range are from the slant paths during the banking of the aircraft during the corners of the box pattern. The measurements between the dashed lines are summarized in Fig. 19.

flight is shown in Fig. 10. Most of the ground track was covered by coastal forest of tall trees covering hills and low (few km high) coastal mountains. The figure also shows a plot of the time-tagged location and altitude. A time series of the measurement results is shown in Fig. 11. It shows the aircraft and ground elevations computed from range vs. time as well as the lidar-measured differential optical depths (DODs, measured from the peak to offline shoulder) and the retrieved values of XCO$_2$. In this and in similar figures, the scattering surface elevation is computed from the aircraft altitude, the off-nadir beam angle and the lidar-measured slant range to the scattering surface. All measurements plotted are for 10 s averages. Figure 12 shows a photograph of a typical surface measured from the aircraft and a summary of the lidar retrieval statistics vs. altitude for the indicated area in Fig. 11. The corresponding measurements from the AVOCET in situ sensor in the spiral are shown as blue lines and as blue dots for the column average from that altitude to the surface.

## 2014 SF2

The 2014 Science Flight 2 (SF2) targeted measurements over a desert region. The location chosen was western edge of the Mojave Desert in California. The ground track and the time-tagged altitude plot are shown in Fig. 13 and show ap-

proach and spiral down over Edwards Air Force Base (AFB). This flight occurred during a period of widespread atmospheric haze at lower altitudes caused by smoke spreading from a wildfire in the nearby Sierra Nevada. A time series segment of lidar measurements from this flight is shown in Fig. 14. This segment contains a spiral-down maneuver. The height-resolved backscatter profile is shown in Fig. 15. It shows a layer of haze from $\sim 4$ km to the surface caused by smoke from the wildfire. The altitude summary of the lidar measurements is also shown, along with measurements from the in situ sensor. The results show there is very good agreement between the XCO$_2$ retrieved from the lidar and that computed from the in situ sensor, despite the significant optical scattering from the thick haze layer.

## 2014 SF3

The 2014 Science Flight 3 (SF3) was a flight to and from Iowa made in the afternoon and evening, respectively. There were also segments during the transit from California to and from Iowa that allowed assessing the lidar's capability to measure horizontal (east–west) gradients in XCO$_2$. Figure 16 shows the ground track of the 2014SF3G1 segment in Colorado and Nebraska, which was during the west-to-east leg of the flight toward Iowa. Figure 17 shows the ground track of the segment 2014SF3G2, in Iowa, Nebraska and Colorado, which was during the return (east-to-west) flight leg toward California.

The time series of the lidar retrievals of XCO$_2$ during these flights legs are shown in Fig. 18. The outbound (west-to-east) leg flew at only one aircraft altitude, but the return leg flew three altitudes. The data points plotted are for lidar retrievals based on 50 s ($\sim 12$ km along track) averaging. Both segments clearly show the gradual decrease of XCO$_2$ caused by increasing growing crop density (and CO$_2$ uptake) toward the eastern end of the flight legs in the Midwestern US, even for the return segment that used three different aircraft altitudes. The solid lines show the XCO$_2$ values computed from the NASA Parameterized Chemistry Transport Model (PCTM) (Kawa et al., 2004) for these tracks and times. Although there are offsets in the average values, there is good agreement between the east–west gradients measured by the lidar on the outbound (SF3G1) flight segment and those computed from the model, as well as for the 6.3 km altitude leg of the return segment (SF3G2).

## 2014 SF5

The 2014 Science Flight 5 targeted XCO$_2$ over growing corn cropland in Iowa in early morning. Figure 19 shows the ground track of the segment of SF5 made over Iowa along with the time-tagged altitude plot. The spiral-down location was centered near West Branch, Iowa. This flight used a three-box pattern flown from lowest to highest altitude, then a spiral-down maneuver made near the West Branch tower.

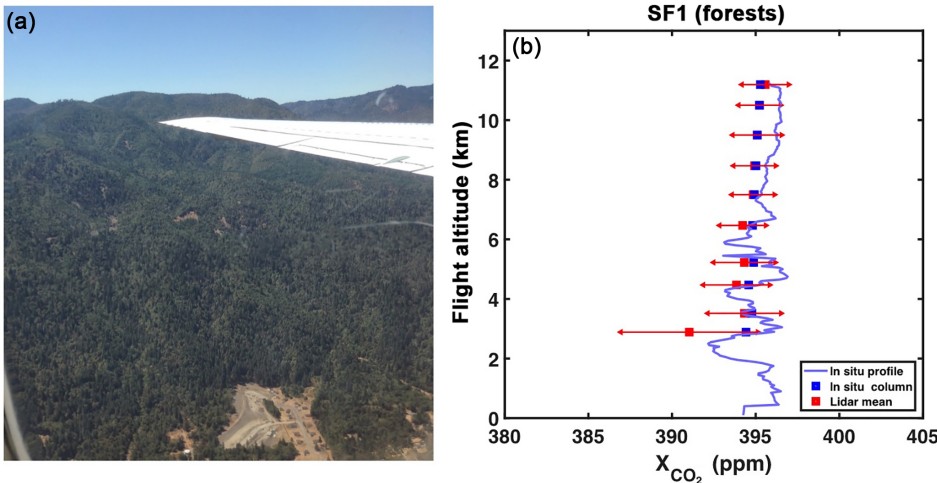

**Figure 12. (a)** Photo of northern CA coastal redwood forests taken from the aircraft on 2014 SF1. **(b)** Summary plot of the in situ (blue) and retrievals from lidar measurements (red) vs. altitude. The lidar results are for $XCO_2$ retrievals based on 10 s average from the altitude where the results are plotted, and the error bars are for 1 SD. The $XCO_2$ computed from the in situ sensor from the plotted altitude to the ground is shown as the blue dots.

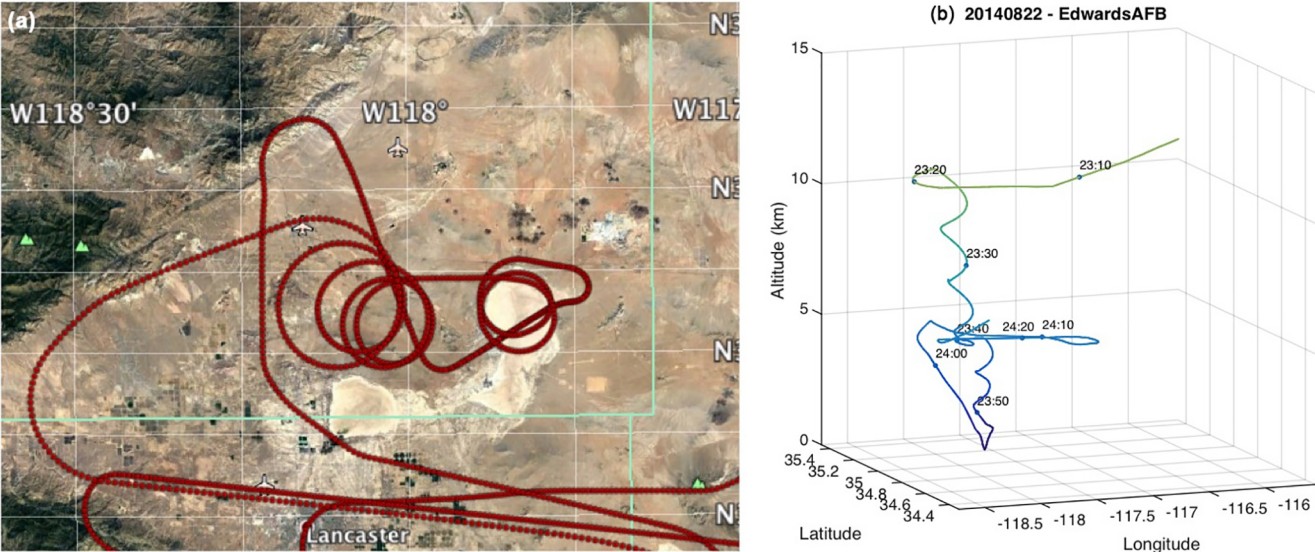

**Figure 13. (a)** Map of the track of the spiral down over Edwards AFB, California, on 2014 SF2 on 22 August 2014. **(b)** Time-tagged location and altitude plot for that flight segment.

Figure 20 shows the time history of the segment of the flight just west of the Rocky Mountains to the box pattern in Iowa. The elevation of the easternmost Rocky Mountains and the longer ranges from the turns in the corners of the box patterns are noticeable in the history. Figure 21 shows a photograph of the Iowa landscape for one leg of the lower box. It also shows the altitude summary for the $XCO_2$ retrievals from the lidar measurements between the dashed lines in Fig. 20. The $XCO_2$ retrievals from the lidar measurements closely follow those from in situ except at the lowest altitude and the gradually increasing values with altitude agree with those com-

puted from the in situ sensor. In the 7–10 km altitude range the retrieved $XCO_2$ for the 2011 flight segment over Iowa had SDs of ∼ 1.8 ppm over Iowa, while in the 2014 flights they were ∼ 1.2 ppm.

## 7 2016 airborne campaign

The 2016 campaign was a short (two flight) campaign flown during the local wintertime. The campaign objective was to assess the performance of the 2016 version of the $CO_2$ Sounder lidar, to assess its measurements made using fewer

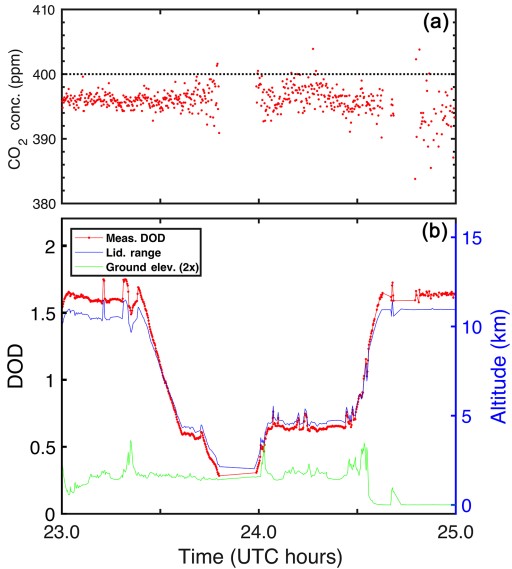

**Figure 14.** Lidar measurement and retrieval results from 2014 SF2 flight over Edwards AFB, CA, on 22 August 2014. From the $\sim$ 11 km altitude, the aircraft flew a spiral-down pattern to near the Edwards Dry Lake Bed. **(a)** The retrieved XCO$_2$ values from the lidar measurements, with each dot made using 10 s averaging time. **(b)** Time-resolved results showing the lidar-measured differential optical depth, the range to the surface and the computed surface elevation. The lidar-measured range to the scattering surface and the scattering surface "ground" elevation are plotted against the right-hand axis. The upward spikes in the DOD and range are from the slant paths during the banking of the aircraft during the corners of the box pattern.

wavelength samples and, with additional laser power, to characterize the measurements made at low sun angles over fresh cold snow. The changes in the instrument from the 2014 version are summarized in Table 1.

## 2016 desert TS4 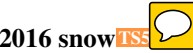

The 2016 desert flight was made again over the Mojave Desert and Edwards AFB, CA, which was used for the spiral-down location. Figure 22 shows a plot of the ground track and the time-tagged altitude plot for the flight. Figure 23 shows the altitude summary of the lidar measurements for the spiral down and their comparison to the in situ measurements. The plot format is the same as for Fig. 13, except that these measurements have 1 s averaging time. The smallest SDs for the 1 s measurements were $\sim 0.7$ ppm for altitudes between 7 and 10 km, which is a factor of $\sim 8$ smaller than corresponding lidar measurements made in 2011. The SD of the lidar 1 s retrievals vs. altitude is also shown in the figure, along with those computed from a statistical model of the lidar (Sun et al., 2017a). The altitude dependence of both plots is quite similar, with the SDs increasing at lower altitudes due to decreasing optical depth of the CO$_2$ line and at

upper altitudes due to the R$^{-2}$ dependence of the lidar signal and the increased attenuation of the stronger CO$_2$ absorption. The plot also shows that the SDs of the retrievals are about a factor of 1.5 higher than the lidar model. After the campaign, investigations found that the detector electronics may have contributed some additional noise. Improvements in the detector electronics were made for the 2017 ASCENDS airborne campaign and the impact on the lidar retrievals will be assessed as part of the data analysis.

## 2016 snow TS5

The 2016 snow flight targeted a long series of measurements over fresh cold snow. Snow had recently fallen in northeast Nevada and the surface temperatures had stayed below 0° C, so the flight repeated a north–south route just south of Elko, NV. The Elko, NV, airport was the nearest location available for the spiral-down maneuvers. The flight altitudes of the north–south legs of this flight were between 6.6 and 9.5 km. Figure 24 shows a plot of the flights ground track and a photograph of the snow-covered desert surface made from the airplane. The altitude summary of the lidar measurements for this flight is shown in Fig. 25. To investigate measurement approaches planned for space we also used three different laser configurations for this flight. Those were 30 sample wavelengths and one laser amplifier, 15 sample wavelengths and one laser amplifier, and 15 wavelengths and two laser amplifiers. The second amplifier almost doubled the transmit power to 50 uJ/pulse. As expected the 30 and 15 wavelength samples with one laser amplifier (same average power) gave similar results. The SDs for two-amplifier lidar setting were also smaller than for one amplifier due to the larger received signal and hence higher SNR. Also, as expected from the snow surface's low ($\sim 4$ %) reflectivity, the measurement SDs over the snow were about 3 times higher than those over the desert.

## 8   Discussion

The flights and height-resolved measurement statistics from the 2014 and 2016 airborne campaigns are summarized in Tables A1 and A2 in the Appendix. All column entries in these tables, except counts and SDs, are the average values for the measurements binned by altitude. The 2014 altitude bins typically averaged 30 of the 10 s measurements, which at a nominal 200 m s$^{-1}$ aircraft speed meant $\sim 60$ km along-track averaging. The 2016 altitude bins typically averaged 150 of the 1 s measurements, resulting in $\sim 30$ km along-track averaging. Plots of the differences between the lidar-measured XCO$_2$ and those computed from the in situ sensor are shown in Fig. 26, along with the number of measurements for the data set and error bars. In the tables, the measurement DOD is computed from the fitted retrieval from the line peak to the line shoulder. The offline lidar total signal column is

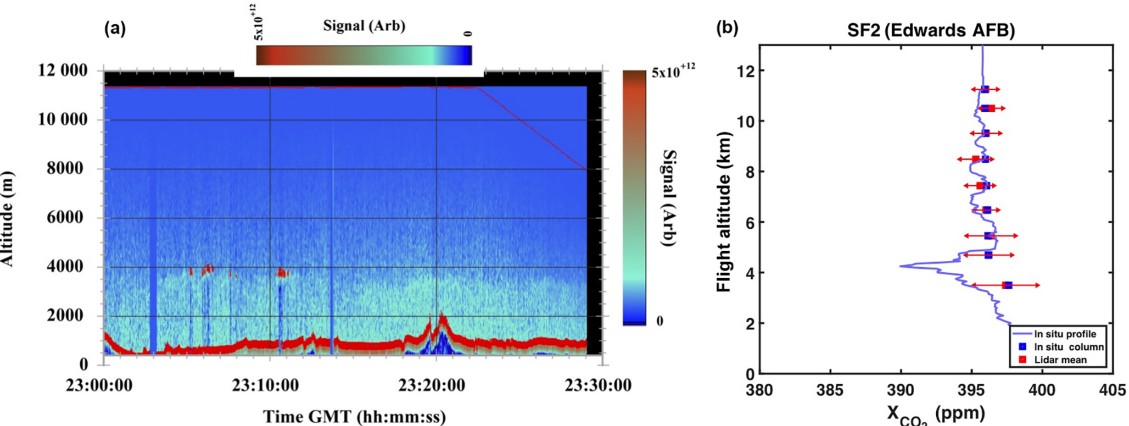

**Figure 15. (a)** Time history of the range-resolved backscatter for the offline wavelengths recorded on 2014 SF2 before the spiral-down maneuver. The plot shows enhanced scattering from haze in the boundary layer. The aircraft altitude is the thin red line at the top of the plot. Each vertical profile is $R^2$ corrected and used 1 s averaging. **(b)** Summary of the in situ (blue) CO$_2$ measurements and the XCO$_2$ retrievals from lidar measurement (red) vs. altitude for the segment in Fig. 15. The lidar results are for XCO$_2$ retrievals based on 10 s average from the altitude where the results are plotted to the surface, and the error bars are for 1 SD. The XCO$_2$ values computed from the in situ sensor from the plotted altitude to the ground are shown as the blue dots.

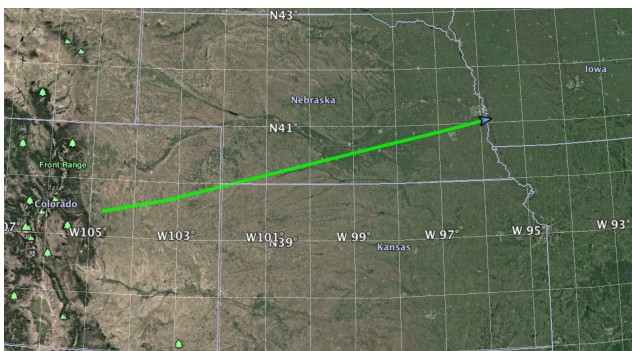

**Figure 16.** Map of the segment of the west-to east flight track analyzed for 2014 SF3 on 25 August 2014 approaching Iowa.

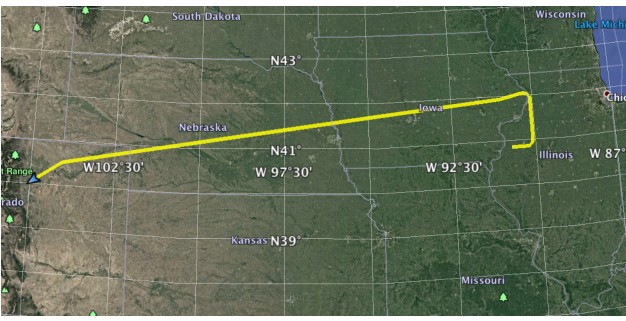

**Figure 17.** Map of the segment of the east-to-west flight track analyzed during 2014 SF3 on 25 August 2014 when leaving Iowa and approaching Colorado.

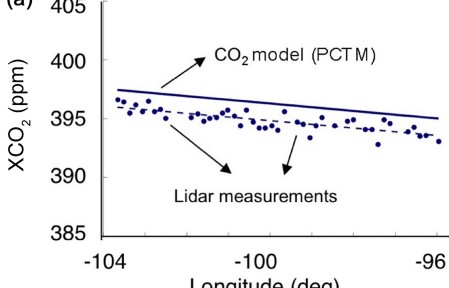

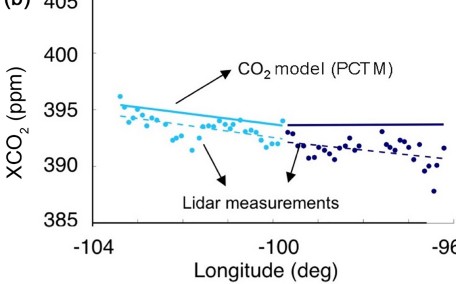

**Figure 18.** Retrieved XCO$_2$ from lidar measurements vs. longitude for the transit flights to (from) Colorado to (from) Iowa on 2014 SF3. **(a)** Outbound leg (west-to-east flight direction, SF3G1, measured from 11.2 km altitude) and **(b)** return flight leg (east-to-west flight direction, SF3G2, with dark blue points measured from 5.6 km altitude and light blue points from 6.3 km altitude). The measurements shown are for retrievals using 50 s data averages, and the flight altitudes are indicated. The solid lines show the XCO$_2$ values computed from the PCTM atmospheric model for that location and these times.

for detected photons per wavelength, summed over the averaging time.

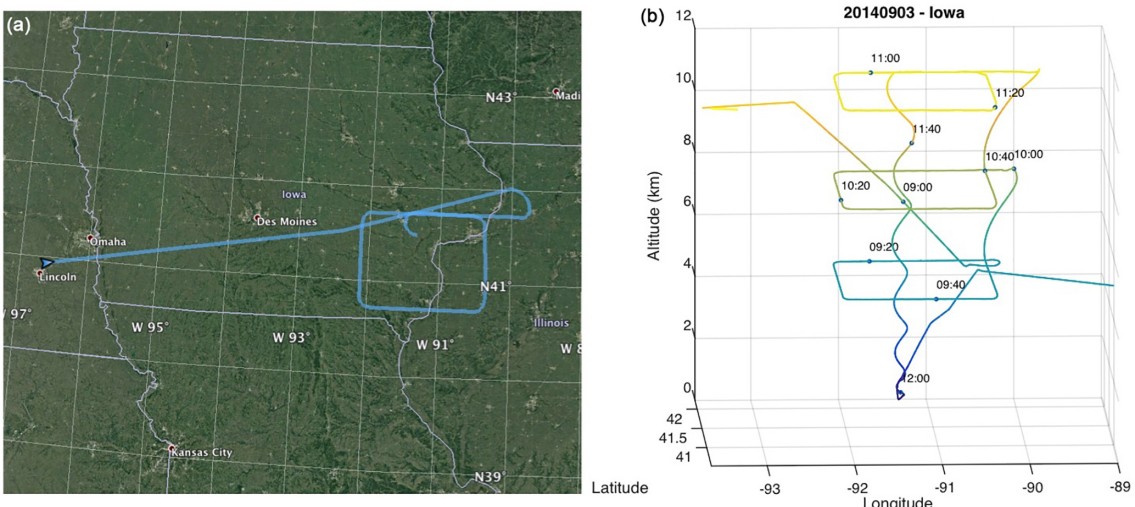

**Figure 19. (a)** Map of the flight track from Lincoln, Nebraska, from west to east, and a box pattern made over Iowa at dawn on the 2014 SF5 flight on 3 September 2014. **(b)** Plot of the time-tagged location and altitude for three-altitude box pattern during the same flight.

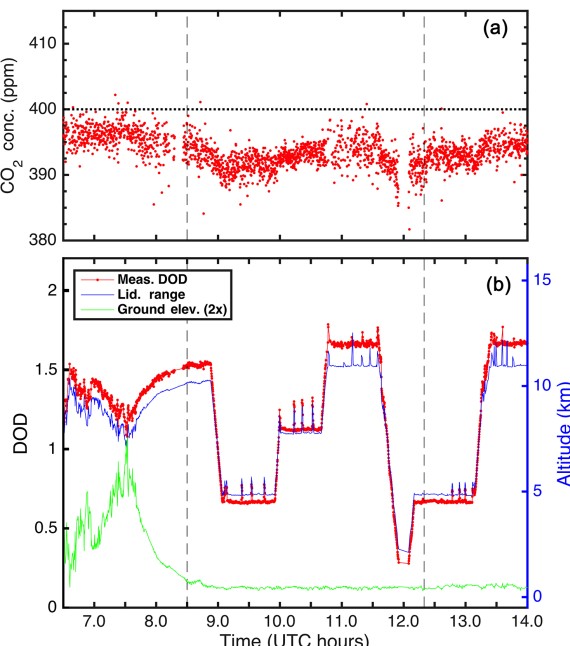

**Figure 20.** Lidar measurement and retrieval results from 2014 SF5 flight over Iowa on 3 September 2014. This flew a square flight pattern near the NOAA West Branch, Iowa, tower at three different altitudes. **(a)** The retrieved XCO$_2$ values from the lidar measurements, with each dot made using 10 s averaging time. **(b)** Time-resolved results showing the lidar-measured differential optical depth, the range to the surface and the computed surface elevation. The lidar-measured range to the scattering surface and the scattering surface "ground" elevation are plotted against the right-hand axis. The upward spikes in the DOD and range are from the slant paths during the banking of the aircraft during the corners of the box pattern. The measurements between the dashed lines are summarized in Fig. 21.

The results show that in 2014, typical SDs in retrievals based on 10 s averaging were ∼ 1 ppm, with lowest SDs over desert and slightly higher values over forest. The lidar changes made for the 2016 reduced the speckle noise and the signal shot noise in the measurements and improved the performance. For the 2016 flights, ∼ 0.7 ppm SDs were achieved over desert with 1 s averaging time, with 2.5 ppm SDs measured over snow surfaces. As was seen in the 2011 airborne measurements (Abshire et al., 2013b) the SDs of the XCO$_2$ retrievals vary with altitude. At lower altitudes the optical depth of the line is smaller, which magnifies the lidar measurement error, and since the received signal varies as R$^{-2}$, at higher altitudes the lower signal levels limit the measurement resolution. As a result there is an altitude with smallest SD, which for the 2016 flights was ∼ 8 km. In all cases the agreement between the lidar-measured XCO$_2$ and that computed from the in situ sensor and MERRA atmospheric model was < 1 ppm.

Two experiments using slightly different lidar transmitter configurations were conducted during the 2016 snow flight. The results show that reducing the number of laser measurement wavelengths from 30 to 15 using the same average laser power had only a minor impact (changed mean XCO$_2$ ∼ 0.5 ppm, increased SD to ∼ 0.4 ppm) on the retrieval results. They also show that adding an additional fiber amplifier to the transmitter to double the laser energy increased the received signal and reduced the measurement SD, as expected.

Since CO$_2$ fluxes make only small changes in the column average, it is important to understand the causes of the differences between XCO$_2$ values calculated from the in situ sensor measurements and those from the lidar retrievals. The laser's wavelength locking was quite good, as was the ranging accuracy, so residual errors from those potential sources are likely small. There are several other potential sources

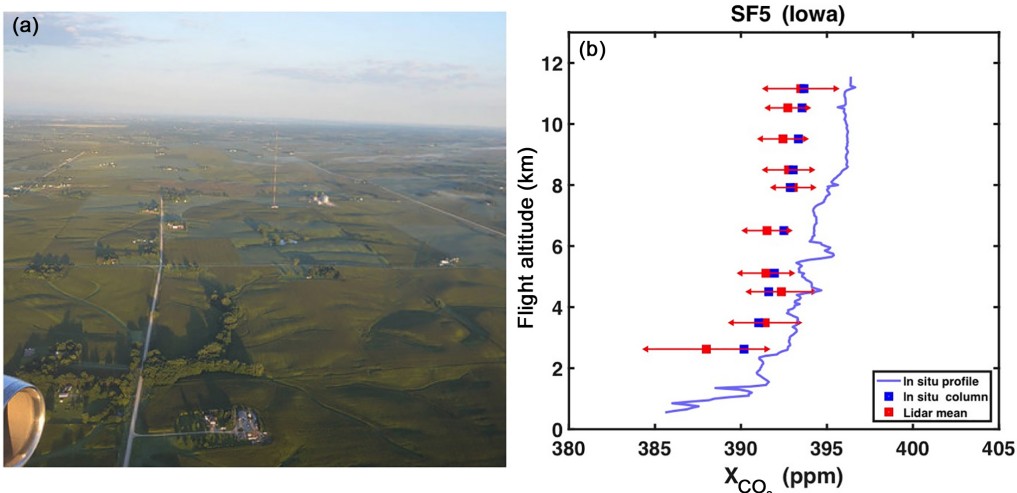

**Figure 21. (a)** Photo of the Iowa topography and the West Branch tower taken from the aircraft on 2014 SF5. **(b)** Summary of the in situ (blue) and the retrievals from the lidar measurement (red) vs. altitude. The lidar results are for retrievals based on 10 s average from the altitude where the results are plotted, and the error bars are for 1 SD. The XCO$_2$ computed from the in situ sensor from the plotted altitude to the ground are shown as the blue dots. The in situ sensor shows the drawdown in CO$_2$ concentrations at lower altitudes caused by cropland, and that general trend is seen in the XCO$_2$ values computed from in situ and in lidar retrievals.

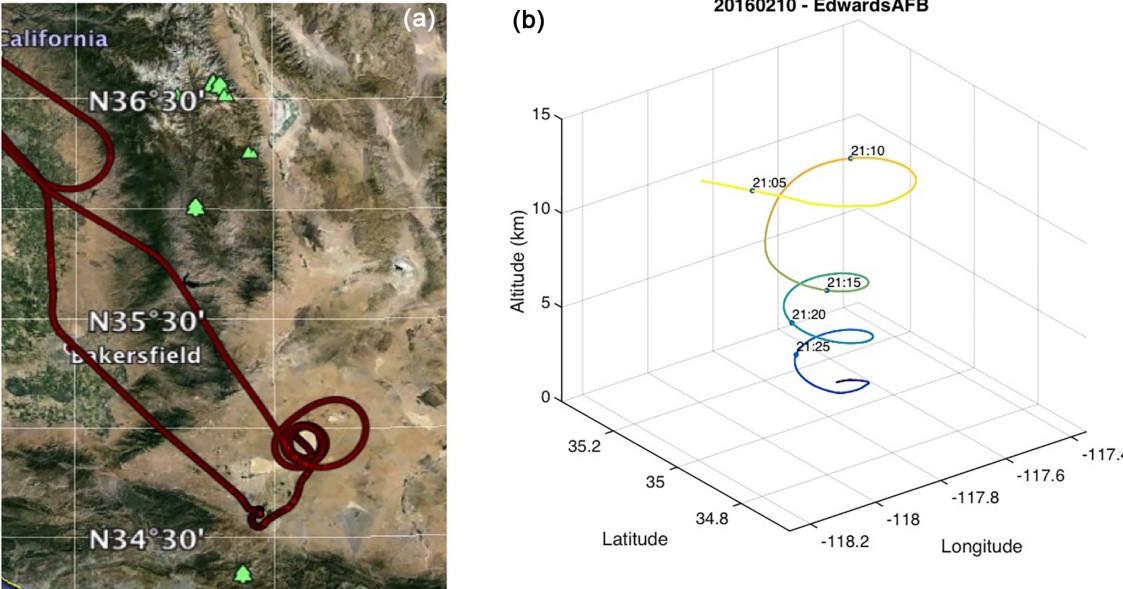

**Figure 22. (a)** Flight track for 10 February 2016 flight over Edwards AFB, California. **(b)** Time-tagged location and altitude plot for the spiral-down maneuver over Edwards AFB for the same flight.

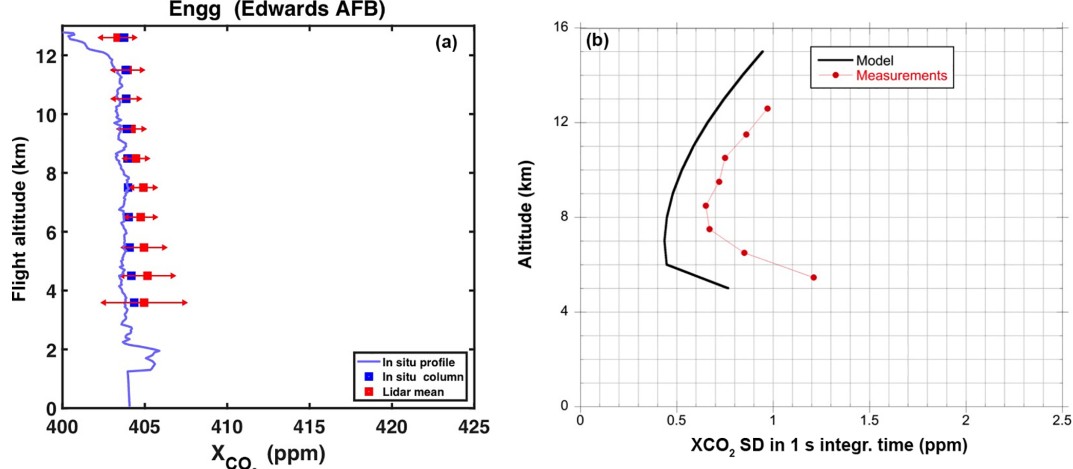

**Figure 23. (a)** Plot of the measurements made during the spiral-down segment of the 2016 desert flight over the Rogers Dry Lake bed near Edwards AFB, CA. The XCO$_2$ retrievals from the lidar measurements are shown (in red) from the plotted altitude to the surface, the in situ CO$_2$ concentration measurements (blue line) and the XCO$_2$ computed from the in situ CO$_2$ readings from the plotted altitude to the surface (blue dots). **(b)** Plot of the SD of the XCO$_2$ retrievals from the lidar measurements (red) using 1 s integration time, showing best resolution near 8 km altitude. The solid black line represents the SDs computed from a statistical model of the lidar measurement.

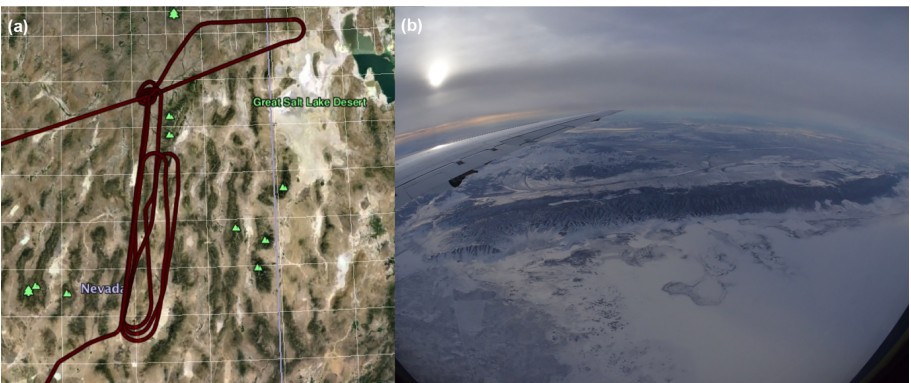

**Figure 24. (a)** Map of the ground track for the 2016 snow flight made over northeastern Nevada. The spiral-down location was centered on the airport at Elko NV and all subsequent measurements were made during the north–south tracks south of Elko. **(b)** Photograph of the snow covered hills and desert floor made during the 2016 snow flight.

for ppm-level differences. Any small slowly moving changes in the lidar's response vs. wavelength or other factors that are not modeled in the retrieval algorithm will cause biases. Previous work (Abshire, 2013b) also showed that the mean retrieved XCO$_2$ values were sensitive, at the few ppm level, to the source of the reference atmosphere (for example, MERRA or that from the DC-8) used for the retrieval's LUTs. If the atmosphere is not in steady state, then the actual CO$_2$ concentrations in the column may be slowly drifting vs. time before, during and after the spiral maneuvers used for comparison. All these potential sources of difference and bias need to be investigated in future work. The recently completed ASCENDS 2017 airborne campaign (Abshire, 2017) has provided a new and extensive data set that can be used for this purpose. It carried out a robust calibration flight that

had 10 spiral maneuvers along with 7 additional flights made under a wide variety of conditions.

Work has been ongoing at NASA Goddard Space Flight Center for several years to extend the airborne CO$_2$ Sounder lidar's measurement capability to orbit for a space mission like ASCENDS (NASA ASCENDS White Paper, 2018). The key capabilities needed are a performance model that allows accurately scaling the characteristics of the airborne measurements to space and the laser and detectors with the needed performance in the space environment. The present plans for a space-based version of this lidar use 16 laser wavelengths. Recent summaries are available on the modeling the space-based lidar measurement performance and on determining the needed laser power (Sun et al., 2017a), on developing the rugged higher power laser (Stephen et al.,

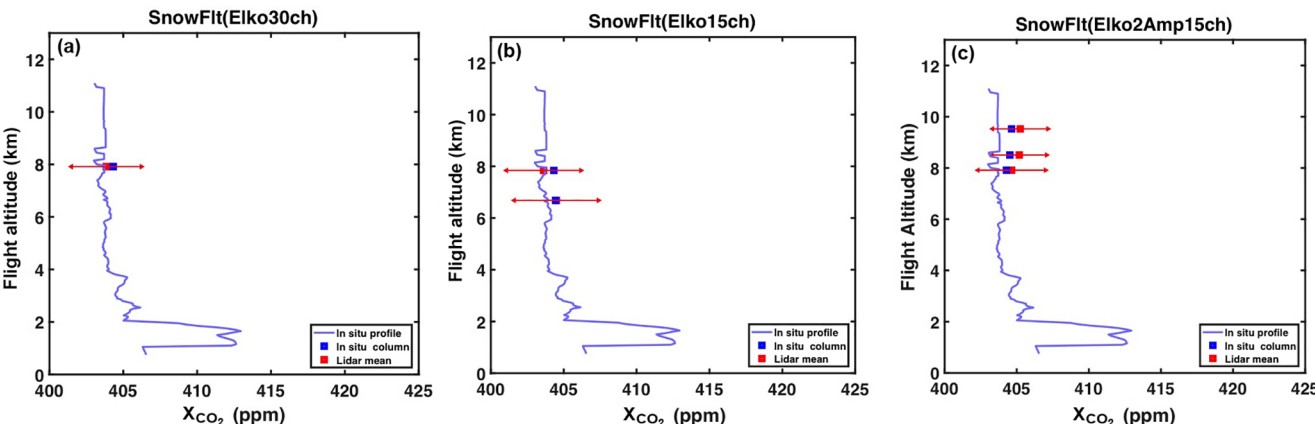

**Figure 25.** Lidar results from the 2016 snow flight. **(a)** Plot of the XCO$_2$ measurements from the lidar (in red) from the plotted altitude to the surface, the in situ CO$_2$ concentration measurements (blue line) and the XCO$_2$ computed from the in situ CO$_2$ readings (blue dots) from the plotted altitude to the surface (blue dots). Here the lidar measurements were made using 30 laser wavelength samples across the CO$_2$ line. **(b)** Results over the same snow area, but lidar measurements were made using 15 wavelength samples across the CO$_2$ line. **(c)** Results over the same snow area, but with lidar measurements made using 15 wavelengths and using two EDFA laser amplifiers in parallel.

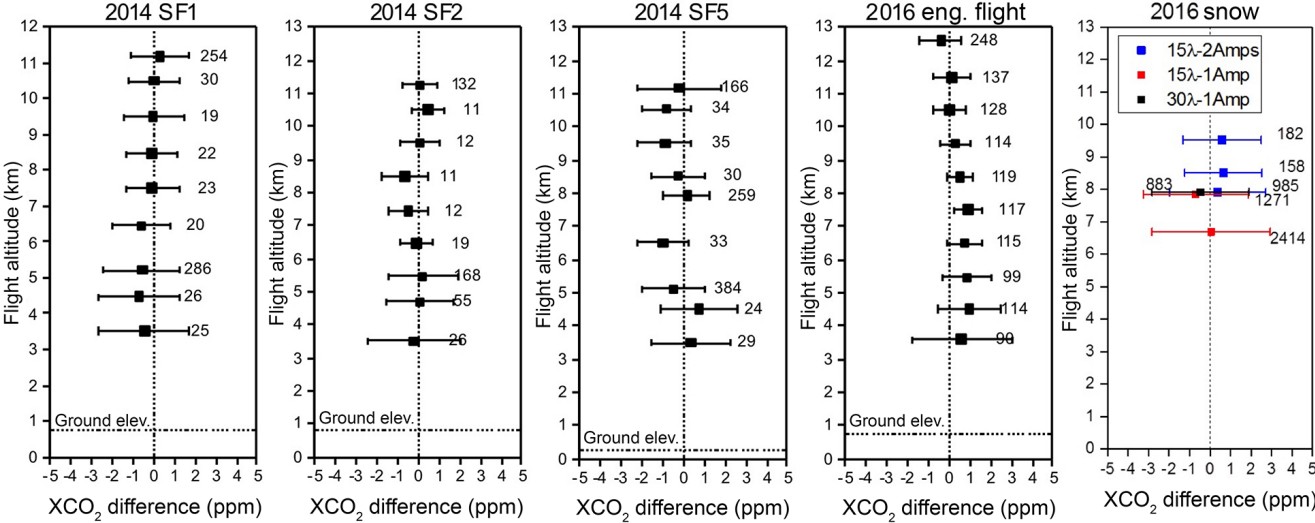

**Figure 26.** Summary of results from the 2014 and 2016 flights, plotted from the values summarized in Tables 3 and 4. The dots are the mean value of the XCO$_2$ from the lidar minus that computed from the in situ sensor. They are plotted at the altitude from which they were measured, and the average ground elevations are also shown. The 2014 statistics are from data using 10 s averaging and the 2016 measurements used 1 s averaging. The error bars are those of the lidar data set, and the numbers shown are the number of lidar observations in that set. There were three different settings used in the lidar for the 2016 snow flight, and their results are plotted in different colors.

2017, 2018; Nicholson et al., 2016) and on developing the HgCdTe APD detector needed in the lidar receiver (Sun et al., 2017b) for a space mission. An engineering model of the receiver's HgCdTe APD detector–cooler assembly has passed space qualification and radiation testing and has the sensitivity needed for a space mission. An engineering model of the space lidar's key electro-optic assemblies will undergo space environmental testing during spring 2018.

## 9 Summary

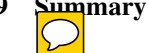

Since its use in the 2011 campaign (Abshire et al., 2013b), our team has made several improvements to the CO$_2$ Sounder airborne lidar. These included incorporating a rapidly wavelength tuneable step-locked seed laser in the lidar transmitter, using a much more sensitive HgCdTe APD detector and using a digitizer with higher measurement rate in the receiver. We also improved the lidar calibration approach, the XCO$_2$ retrieval algorithm and the approach used to minimize the

impact from a nearby isotopic water vapor (HDO) line. In 2016 we used a larger laser divergence angle and improved the transmission of the receiver optics and the uniformity of the illumination pattern on the detector pixels. All these changes considerably improved the lidar's precision, stability and accuracy.

The improved CO$_2$ Sounder lidar was used to make measurements during the ASCENDS 2014 and 2016 airborne campaigns. These were made over several types of surfaces from 3 to 12 km aircraft altitudes. The results are compared to the XCO$_2$ values computed from an airborne in situ sensor during spiral-down maneuvers. The 2014 results also show measurement of horizontal gradients in XCO$_2$ made over the Midwestern US on two flight segments that were consistent with those computed from a chemistry transport model. Analysis show the 2014 and 2016 measurements have consistent agreements within 1 ppm for mean value of XCO$_2$ compared to that computed from the in situ sensor, which is better than those for 2011 version.

Retrievals for the 2016 airborne lidar measurements made over desert surfaces show precision of 0.8 ppm with 1 s averaging, which is ~3 times smaller than similar measurements made in 2011. In 2016 measurements were also made over fresh snow surfaces, which are important for high-latitude studies, but which are dark at CO$_2$ measurement wavelengths. As expected, the SDs of lidar-measured XCO$_2$ were about 3 times larger over snow surfaces. Over snow the agreements of the mean values of lidar retrievals with XCO$_2$ computed from the in situ sensor were also within 1 ppm. The 2016 lidar's precision and consistent sub-ppm agreement with the XCO$_2$ calculated from in situ sensors are expected to benefit future airborne carbon science campaigns. They also help advance the technique's readiness for a future space-based instrument.

*Data availability.* All of the data used in this work are available from the first author (james.b.abshire@nasa.gov).

## Appendix A

Tables A1 and A2 summarize the measurement statistics for five flights made during the 2014 and 2016 campaigns.

**Table A1.** Summary of results for three of the 2014 flights.

| Lidar and measurement conditions | Aircraft altitude (km) | Ground elevation (km) | Mean slant range (km) | No. of meas. in alt bin | Lidar DOD | Lidar offline tot. signal (K counts) | Lidar XCO$_2$ mean (ppm) | Lidar XCO$_2$ SD (ppm) | AVOCET XCO$_2$ (ppm) | XCO$_2$ difference: (mean lidar – AVOCET) (ppm) |
|---|---|---|---|---|---|---|---|---|---|---|
| SF1 | 3.518 | 0.752 | 2.861 | 25 | 0.384 | 5590 | 394.32 | 2.2 | 394.78 | −0.45 |
| Forests | 4.469 | 0.814 | 3.773 | 26 | 0.516 | 3474.4 | 393.85 | 1.99 | 394.58 | −0.73 |
| 30 wavelengths | 5.224 | 0.756 | 4.489 | 286 | 0.623 | 1993 | 394.32 | 1.85 | 394.88 | −0.56 |
| 1 amplifier | 6.467 | 0.799 | 5.822 | 20 | 0.826 | 1222.6 | 394.22 | 1.39 | 394.82 | −0.6 |
| 10 s average | 7.498 | 0.722 | 6.988 | 23 | 1.008 | 802.2 | 394.85 | 1.28 | 394.93 | −0.08 |
| | 8.468 | 0.596 | 8.361 | 22 | 1.222 | 624.7 | 394.93 | 1.26 | 395.02 | −0.08 |
| | 9.5 | 0.865 | 8.934 | 19 | 1.333 | 531.1 | 395.08 | 1.47 | 395.1 | −0.01 |
| | 10.502 | 0.644 | 10.23 | 30 | 1.543 | 313.6 | 395.23 | 1.26 | 395.22 | 0.01 |
| | 11.193 | 0.757 | 10.582 | 254 | 1.613 | 395 | 395.57 | 1.4 | 395.27 | 0.3 |
| | 3.498 | 0.747 | 2.929 | 26 | 0.392 | 13 304 | 397.4 | 2.23 | 397.62 | −0.22 |
| SF2 | 4.692 | 0.842 | 4.125 | 55 | 0.565 | 6156.1 | 396.22 | 1.62 | 396.17 | 0.04 |
| Desert | 5.454 | 0.859 | 4.74 | 168 | 0.658 | 3837.1 | 396.37 | 1.71 | 396.15 | 0.21 |
| 30 wavelengths | 6.473 | 0.914 | 5.925 | 19 | 0.839 | 2506.4 | 396.02 | 0.83 | 396.12 | −0.11 |
| 1 amplifier | 7.438 | 0.824 | 7.236 | 12 | 1.038 | 1759.2 | 395.59 | 0.96 | 396.06 | −0.46 |
| 10 s average | 8.484 | 0.842 | 8.393 | 11 | 1.227 | 1299.1 | 395.29 | 1.12 | 395.98 | −0.68 |
| | 9.51 | 0.843 | 9.513 | 12 | 1.415 | 937.2 | 396.03 | 0.96 | 395.95 | 0.07 |
| | 10.497 | 0.815 | 10.539 | 11 | 1.592 | 824.1 | 396.4 | 0.79 | 395.94 | 0.46 |
| | 11.246 | 0.78 | 10.604 | 132 | 1.614 | 663.2 | 395.99 | 0.83 | 395.93 | 0.06 |
| SF5 | 3.489 | 0.228 | 3.425 | 29 | 0.453 | 4713.1 | 391.4 | 1.94 | 391.03 | 0.37 |
| Cropland | 4.504 | 0.22 | 4.455 | 24 | 0.604 | 2763.4 | 392.35 | 1.86 | 391.61 | 0.74 |
| 30 wavelengths | 5.112 | 0.219 | 4.96 | 384 | 0.682 | 2038.9 | 391.45 | 1.48 | 391.95 | −0.49 |
| 1 amplifier | 6.507 | 0.225 | 6.533 | 33 | 0.922 | 1043 | 391.52 | 1.26 | 392.5 | −0.98 |
| 10 s average | 7.92 | 0.22 | 7.834 | 259 | 1.134 | 741.2 | 393.04 | 1.13 | 392.87 | 0.17 |
| | 8.499 | 0.224 | 8.554 | 30 | 1.249 | 582.3 | 392.77 | 1.33 | 393.04 | −0.27 |
| | 9.513 | 0.227 | 9.51 | 35 | 1.409 | 423.4 | 392.44 | 1.28 | 393.34 | −0.9 |
| | 10.529 | 0.238 | 10.569 | 34 | 1.59 | 293 | 392.72 | 1.13 | 393.56 | −0.83 |
| | 11.158 | 0.218 | 10.974 | 166 | 1.665 | 136.9 | 393.48 | 2.03 | 393.67 | −0.19 |

**Table A2.** Summary of results for the 2016 flights.

| Lidar and measurement conditions | Aircraft altitude (km) | Ground elevation (km) | Mean slant range (km) | No. of meas. in alt bin | Lidar DOD | Lidar offline tot. signal (K counts) | Lidar XCO$_2$ mean (ppm) | Lidar XCO$_2$ SD (ppm) | AVOCET XCO$_2$ (ppm) | XCO$_2$ difference: (mean lidar – AVOCET) (ppm) |
|---|---|---|---|---|---|---|---|---|---|---|
| Engineering | 3.589 | 0.734 | 2.997 | 90 | 0.424 | 2690 | 404.96 | 2.43 | 404.36 | 0.6 |
| Desert | 4.503 | 0.726 | 3.965 | 114 | 0.572 | 1731.5 | 405.16 | 1.52 | 404.18 | 0.98 |
| 30 wavelengths | 5.464 | 0.715 | 4.968 | 99 | 0.73 | 1726.8 | 404.95 | 1.21 | 404.08 | 0.87 |
| 1 amplifier | 6.496 | 0.76 | 6.005 | 115 | 0.902 | 2729.4 | 404.75 | 0.85 | 404.02 | 0.73 |
| 1 s average | 7.501 | 0.724 | 7.078 | 117 | 1.082 | 2031.8 | 404.91 | 0.68 | 403.98 | 0.93 |
| | 8.49 | 0.714 | 8.154 | 119 | 1.267 | 1605.3 | 404.46 | 0.65 | 403.95 | 0.51 |
| | 9.495 | 0.799 | 9.092 | 114 | 1.437 | 1268 | 404.19 | 0.72 | 403.91 | 0.28 |
| | 10.519 | 0.798 | 9.843 | 128 | 1.58 | 1122.4 | 403.87 | 0.75 | 403.86 | 0.01 |
| | 11.497 | 0.816 | 11.304 | 137 | 1.845 | 876.9 | 403.95 | 0.86 | 403.84 | 0.11 |
| | 12.601 | 0.795 | 12.227 | 248 | 2.022 | 710.5 | 403.34 | 1 | 403.74 | −0.4 |
| Snow 30 wavelengths 1 amplifier 1 s average | 7.914 | 1.848 | 6.081 | 883 | 0.945 | 394.2 | 403.85 | 2.37 | 404.32 | −0.47 |
| Snow 15 wavelengths 1 amplifier 1 s average | 6.682 | 1.907 | 4.805 | 2414 | 0.73 | 1255.6 | 404.49 | 2.89 | 404.45 | 0.05 |
| | 7.841 | 1.875 | 6.008 | 1271 | 0.932 | 827.3 | 403.62 | 2.56 | 404.32 | −0.7 |
| Snow 15 wavelengths 2 amplifiers 1 s average | 7.912 | 1.902 | 6.026 | 985 | 0.94 | 1185.5 | 404.65 | 2.35 | 404.28 | 0.37 |
| | 8.506 | 1.64 | 7.166 | 158 | 1.128 | 1066.5 | 405.18 | 1.89 | 404.52 | 0.65 |
| | 9.525 | 1.454 | 8.19 | 182 | 1.307 | 632.9 | 405.24 | 1.92 | 404.64 | 0.59 |

*Competing interests.* The authors declare that they have no conflict of interest.

*Acknowledgements.* We are grateful to NASA ASCENDS mission's Pre-formulation Activity for supporting the airborne campaigns and to NASA's Earth Science Technology Office and the NASA Goddard IRAD program for supporting the improvements to the CO$_2$ Sounder lidar. We also are grateful for the leadership of Edward V. Browell on the 2014 ASCENDS campaign and appreciate the collaborations with the NASA LaRC's AVOCET and MFFL teams and the JPL LAS team. We also appreciate the support of Frank Cutler, Tim Moes and the DC-8 aircraft team at NASA's Armstrong Flight Research Center on both campaigns. We are also grateful to the reviewers for their detailed and helpful suggestions.

Edited by: William R. Simpson
Reviewed by: two anonymous referees

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

**Remarks from the language copy-editor**

CE1 Grammatically, "is undergoing" is incorrect, as it is the present continuous and the testing is not occurring at the moment (or more accurately, will not be occurring at any given time of reading). The most accurate would be "will undergo" to indicate the future at the time of publication. Alternatively, we could write "will be undergoing".

**Remarks from the typesetter**

TS1 Please confirm.
TS2 Please confirm.
TS3 Please confirm sectioning.
TS4 This is lowercase because the number, 2016, is the beginning of the sentence.
TS5 This is lowercase because the number, 2016, is the beginning of the sentence.
TS6 Please confirm.
TS7 Please confirm.
TS8 Please confirm.