# Peer review of "Airborne Measurements of CO2 Column Concentrations made with a Pulsed IPDA Lidar using a Multiple-Wavelength-Locked Laser and HgCdTe APD Detector"

_Atmospheric Measurement Techniques, 2017_

## Referee Comment (RC1) · Anonymous Referee #1 · 4 Dec 2017

This is a descriptive overview manuscript on the topic of the NASA Goddard CO2 Sounder lidar, the retrieval algorithm, and associated data product examples from two airborne campaigns. These campaigns each consisted of multiple flights and took place in the years 2014 and 2016. The accompanying figures (photos, graphics) are informative. In general it's a well-organized paper that should be of interest to a large segment of the readers. Here are a few comments and questions relating to spots where clarification would be helpful.

(1) Section 3, line 40: Here reference is made to the optical bandpass filters and accompanying Figure 5. It isn't clear why the 2014 filter was not used in the 2016 flights. The 2016 filter appears to be worse in terms of variability of transmission vs. wavelength.

(2) Section 4: Figure 8 is an informative diagram of the retrieval algorithm. It is stated on the figure and on page 5, line 38, that the initial computation assumes a vertically uniform mixing ratio. Later, page 6, line 13, in the paragraph devoted to "clumped fitting", it is stated that clumped fitting "solves for terms like the XCO2 vertical gradient.." Please expand on how this is accomplished.

(3) Section 4: The clumped fitting appears to be an important step, mitigating several potential sources of bias. It would be helpful to expand on this. Also, can you cite one or more references relevant to the multi-pixel approach used by OCO-2 or AIRS?

(4) Section 6 and Figure 11: Figure 11, cited on page 5, line 46, is difficult to interpret. First, it would help to enlarge it. Also, it's not clear from the legend and the plots which is the lidar range and which is the ground elevation. What is the source of the ground elevation numbers? Is it ground elevation or scattering surface elevation from the altimetry? Since SF1 is over the redwoods, this is a case where the potential capability of the lidar to distinguish the optical scattering surface elevation from the ground elevation should become apparent, provided that accurate measurements can be made over short distance scales.

(5) Table 3: This table lists a few numbers from "mean slant range" and "ground elevation" These appear to be averages over each of the fixed altitude segments. As such, they are averages over several km. Correct? What do we learn from these numbers?

---

## Referee Comment (RC2) · Anonymous Referee #2 · 19 Dec 2017

**Airborne Measurements of CO$_2$ Column Concentrations made with a Pulsed IPDA Lidar using a Multiple-Wavelength-Locked Laser and HgCdTe APD Detector**

Atmos. Meas. Tech. Discuss., https://doi.org/10.5194/amt-2017-360

This paper successfully details the results of the XCO2 dataset produced by the CO2 Sounder lidar during its 2014 and 2016 ASCENDS flights. It does a nice job of detailing the motivation and science goals of the ASCENDS mission, the CO2 Sounder lidar's role in that mission, the current state of lidar technology, and the CO2 Sounder's performance in the 2014 and 2016 flights. The paper is readable, and flows well, even in more technical sections such as the instrument setup and retrieval algorithm description. The included figures are, overall, very informative and helpful.

However, I would argue that the paper currently lacks a scientific punch. Providing more context for the instrument performance would help better communicate the scientific and technological impact of the results. Placing stronger emphasis on past instrument performance, and thus on the improvements in data quality since 2011, would help readers appreciate the instrument's achievements. For example, specific comparisons of 2014/16 noise or in situ bias to 2011 flight legs of similar altitudes; to 2011 spirals. Repetition of specific precision and accuracy improvements throughout the text can only improve readers' impression of the instrument progress.

*General comments:*

While the paper does a good job of giving the broad science background for the instrument, a background on past instrument performance would lend important context to the quality of the results. One way to achieve this would be to include a paragraph on past campaign data in the technical background section (section 2). This could also be its own section between sections 2 and 3.

There are several places throughout the text where comparing the 2014 and 2016 data to earlier data would help emphasize the how effective the instrument improvements have been. The abstract, for example, should mention the "five-fold improvement in precision over measurements made in 2011." This is likely the most important take-away message in the entire paper, but is not even given in the abstract. (And, it should be noted, perhaps this number would have be 7.5x better, instead of 5, if not for the "noisy reference voltage"?) This statistic packs a punch, and would draw readers in. Remarking *on* the feasibility of the CO2 Sounder lidar as a future space-based instrument in the abstract, as well, rather than just saying that this study serves as a demonstration of its feasibility, would be a good lead-in for potential readers. How close are you, in some kind of phase space, to technical readiness for a space-based version? Are you there on detectors, or are you a factor of 10 away? Are you there on laser power, or are you a factor of ten away? Throughout the rest of the text, I might recommend pulling numbers from earlier flights' similar-altitude flight legs to compare to those given here for the 2014 and 2016 flights. In the specific comments I list a few places where this can be done – including the summary.

One other area which might benefit from further detail is the 2016 flight that used 15

wavelengths instead of 30 – the authors could explain that a space-based instrument will likely sample fewer wavelengths, so it is important to test the capabilities of the instrument with that change implemented. This would tie this data, once again, to the larger science goal. Speculation on why that sampling change may or may not make a difference in the results, and whether a change was expected, could also be given. The results of other flights using fewer than 30 wavelengths could also be mentioned – did they show any significant difference? It says later that a space-based instrument would only use 8. Why then are you testing with 15 or 30, and not 8?

Finally, more discussion of your ability to determine your biases is needed. A low bias is perhaps the most important potential advantage of the lidar over passive measurements. Can you get your biases down to 0.3 ppm? 0.1 ppm? Therefore, doing a good job to quantify the level at which you can determine your biases is *a critical aspect* to these test flights. It can make or break a decision to do ASCENDS or not. From this paper, it appears that you cannot see any statistically significant biases in your retrieval (as compared to the in-situ). But what is the current level at which you can determine your biases? I note that while you are often within the standard deviation of the different lidar column measurements at a given altitude, you are virtually never within 1 *standard deviation of the mean*. If your errors are truly random, the errors should integrate down like the square root of N. Since you often have ~100 measurements per altitude bin, your noise-driven error is on the order 0.05-0.1 ppm on the mean XCO2 in an altitude bin, while your difference to the in-situ is often more like 0.5 ppm. That is a 5-to-10 sigma discrepancy (ie, it looks like a bias!). However, it could be that the in-situ columns are only good to 0.5 ppm, considering the change in the column that determines over the course of the spiral, the native in-situ errors, etc. Therefore, more commentary on and quantification of your biases and bias knowledge is sorely needed.

*Specific comments:*

Abstract: As stated above, please mention the factor of five improvement in precision in the 2016 flights compared to the 2011 version of the instrument.

Section 1. Intro - When describing currently operational (passive) space-based instruments and their role in carbon cycle science, it might be informative to comment on their abilities to meet the sub-ppm measurement requirement. It would further tie the statistical results of this study to the larger science goal, which is in part to improve upon passive datasets.

Section 2. Instrument background - Paragraph 3 of this section is the only place where the writing feels a bit disjointed and repetitive, and could be reorganized slightly to convey its message more clearly. It hops back and forth from spectroscopic information to environmental variables and Doppler-shift information a few times; I think that a simple rearrangement would help.

Section 3. 2014/16 Lidar setup - it is not clear in the text why the receiver optical transmission (in Table 1, which is referenced on page 3, line 45) was so low in 2014 (9.2%, as compared to ~50% and 60% in 2011, 2016). The authors should comment on this in this section. What specifically was done to improve it so much from 2014 to 2016?

Section 4. Data processing & retrieval – The retrieval summary mentions that "HDO absorption spectrum can bias the retrieved $XCO_2$ value if not taken into account," but it does not give the average value of that bias. The authors would should include either that number or a reference to further information on the topic. Also, there seems to be no reference provided for a more detailed description of the retrieval process. Finally, what is the posterior uncertainty on your measured HDO column? This could be a side benefit of your technique – knowledge of HDO can be used to study water cycle processes (see e.g. Frankenberg et al, 2013, "Water vapor isotopologue retrievals from high-resolution GOSAT shortwave infrared spectra", AMT).

Also, do you fit for a scale factor to a prior CO2 profile, or do you assume a uniform CO2 profile and solve for that single concentration?

Finally, and perhaps most importantly, is the short paragraph on the "clumped fitting" approach. This appears to be a novel aspect to this work. Has this approach of yours been described elsewhere in the literature (if so, please give the reference) or is it introduced here? How much does it reduce the errors (and in what way? Ie – is it simply a lowered scatter at a given altitude level?) between L2a and L2b? And how physically does it reduce the errors – what are the mechanisms? This could be an important aspect to this work, but it is downplayed – not mentioned at all in either the abstract or summary. Is is something that none of the passive teams (or other active teams, to my knowledge) currently do – you figured it out, so please make some more noise about it if it is really useful!

Section 5. Campaigns overview – There are a few opportunities here to reference the precision and accuracy of previous campaign data. For example, on page 6, line 35, the authors could state that this <1ppm agreement between lidar and in situ is improved from 2011 comparisons (Abshire et al. 2013b), when this agreement was more along the lines of <1.4ppm. Also, more should be stated about the "noisy reference voltage". Does this imply that if this were fixed, the precision would be another 33% better? That's significant.

Section 6. I might contest, on the return leg (bottom of Figure 18), the statement "There is good agreement between E-W gradients measured by lidar and those computed from model", especially in the 5.6km segment. The model predicts something like a 1ppm E-W gradient, whereas the observational gradient might be closer to ~4ppm. Please soften this statement.

SECTION 8. Discussion - Line 2: "Changing from 30 to 15 lines did not significantly change the retrievals, per the 2016 snow flight." It wouldn't hurt to include the numbers on this – how much of a difference did it make? Or was it truly negligible?

SECTION 9. Summary – to restate what I said above, more quantification of your biases, and at what level you can even determine them with comparisons to in-situ, is needed.

On line 28 would be meaningful to state specifically how much smaller the lidar vs. in situ biases are than in 2011, and how this was

*Typos/grammar:*
- Page 2, Line 4 – Cleaner wording – remove "so can" from "[…], and so can cause large

retrieval errors."

- Page 2, Line 22 – Define IPDA acronym once before using it in the rest of the text.
- Page 2, Line 42 – Missing a period after "et al" in "[…] Abshire et al, 2013b)."
- Page 5, Line 25 – Typo – "from based on" – remove either "from" or "based on".
- Figure 7 – This figure appears blurry; update resolution.
- Figure 9 – Change RMS labels to include "initial" and "after fit" for better at-a-glance clarity. Also would recommend, in the caption, describing figures in the order in which they appear - top then bottom.
- Figure 14 & similar figures – If range and ground elevation are on the same axis (right-hand), perhaps clarify this by making them the same color but making one dotted/dashed.
- Figure 18
  - The bottom plot doesn't show any 10.8km data, even though it's included in the legend. Is this a mistake? If so, correct plot; if not, removed 10.8km data from legend.
  - A secondary legend for observations vs. model would be better than using arrows.
  - The statistics in the legends don't seem to be described anywhere; describe in caption. Is the correlation between observations and model?
- Page 8, Line 7 – Reference in wrong format? ("[54]")

---

## Author Comment (AC1) · 8 Feb 2018

Thank you for the review and helpful comments. Please find our response along with a revised version of the manuscript, with the changes highlighted, in the attached supplement.

Please also note the supplement to this comment:
https://www.atmos-meas-tech-discuss.net/amt-2017-360/amt-2017-360-AC1-supplement.pdf

---

## Author Response (AR1)

Response to Reviewer Comments for the manuscript:
"Airborne Measurements of $CO_2$ Column Concentrations made with a Pulsed IPDA Lidar using a
Multiple-Wavelength-Locked Laser and HgCdTe APD Detector, "
by James B Abshire et al.
Manuscript #: AMT 2017-360

February 8, 2018

Response to Reviewer 1:

Thank you for the careful review of the manuscript and for your helpful comments. We have updated the manuscript to address all your comments below. In the attached version of the manuscript for the reviewers, all the changes made are highlighted in yellow. I have briefly addressed each comment below as well.

*Referee #1 comments:*
*This is a descriptive overview manuscript on the topic of the NASA Goddard CO2 Sounder lidar, the retrieval algorithm, and associated data product examples from two airborne campaigns. These campaigns each consisted of multiple flights and took place in the years 2014 and 2016. The accompanying figures (photos, graphics) are informative. In general it's a well-organized paper that should be of interest to a large segment of the readers. Here are a few comments and questions relating to spots where clarification would be helpful.*

*1.1 Section 3, line 40: Here reference is made to the optical bandpass filters and accompanying Figure 5. It isn't clear why the 2014 filter was not used in the 2016 flights. The 2016 filter appears to be worse in terms of variability of transmission vs. wavelength.*

The text was updated to address question.

*1.2 Section 4: Figure 8 is an informative diagram of the retrieval algorithm. It is stated on the figure and on page 5, line 38, that the initial computation assumes a vertically uniform mixing ratio.*

No action was requested.

*1.3 Later, page 6, line 13, in the paragraph devoted to "clumped fitting", it is stated that clumped fitting "solves for terms like the XCO2 vertical gradient." Please expand on how this is accomplished.*

*1.4 Section 4: The clumped fitting appears to be an important step, mitigating several potential sources of bias. It would be helpful to expand on this.*

*Also, can you cite one or more references relevant to the multi-pixel approach used by OCO-2 or AIRS?*

Added a subsection on clumped fitting, with the requested additional references.

*1.5 Section 6 and Figure 11: Figure 11, cited on page 5, line 46, is difficult to interpret.*
*1.5 a First, it would help to enlarge it.* The figure was enlarged.

*1.5b Also, it's not clear from the legend and the plots which is the lidar range and which is the ground elevation.* The labels were clarified.

*1.5c What is the source of the ground elevation numbers? Is it ground elevation or scattering surface elevation from the altimetry? Since SF1 is over the redwoods, this is a case where the potential capability of the lidar to distinguish the optical scattering surface elevation from the ground elevation should become apparent, provided that accurate measurements can be made over short distance scales.*

This was clarified in the text.

*1.6 Table 3: This table lists a few numbers from "mean slant range" and "ground elevation"*
*These appear to be averages over each of the fixed altitude segments. As such, they are averages over several km. Correct? What do we learn from these numbers?*

The text was updated with the explanation.

Response to Reviewer Comments for the manuscript:
"Airborne Measurements of $CO_2$ Column Concentrations made with a Pulsed IPDA Lidar using a
Multiple-Wavelength-Locked Laser and HgCdTe APD Detector, "
by James B Abshire et al.
Manuscript #: AMT 2017-360

February 8, 2018

Response to Reviewer 2:

Thank you for the careful and detailed review of the manuscript and for your many helpful comments. We have updated the manuscript to address all your comments below. We feel they have substantially improved the manuscript. In the attached version of the manuscript for the reviewers, all the changes made are highlighted in yellow. I have briefly addressed each comment below as well.

*Reviewer 2 comments:*
*This paper successfully details the results of the XCO2 dataset produced by the CO2 Sounder lidar during its 2014 and 2016 ASCENDS flights. It does a nice job of detailing the motivation and science goals of the ASCENDS mission, the CO2 Sounder lidar's role in that mission, the current state of lidar technology, and the CO2 Sounder's performance in the 2014 and 2016 flights. The paper is readable, and flows well, even in more technical sections such as the instrument setup and retrieval algorithm description. The included figures are, overall, very informative and helpful.*

*2.1a However, I would argue that the paper currently lacks a scientific punch. Providing more context for the instrument performance would help better communicate the scientific and technological impact of the results.*

This was added in several different places as recommended in the detailed comments.

*2.1b Placing stronger emphasis on past instrument performance, and thus on the improvements in data quality since 2011, would help readers appreciate the instrument's achievements. For example, specific comparisons of 2014/16 noise or in situ bias to 2011 flight legs of similar altitudes; to 2011 spirals.*

This was added in several different places as recommended in the detailed comments.

*2.1c Repetition of specific precision and accuracy improvements throughout the text can only improve readers' impression of the instrument progress.*

These were added in several different places as recommended in the detailed comments.

*General comments:*
*2.2 While the paper does a good job of giving the broad science background for the instrument, a background on past instrument performance would lend important context to the quality of the results. One way to achieve this would be to include a paragraph on past campaign data in the technical background section (section 2). This could also be its own section between sections 2 and 3.*

This was added at the end of Section 2.

*2.3 There are several places throughout the text where comparing the 2014 and 2016 data to earlier data would help emphasize the how effective the instrument improvements have been. The abstract, for example, should mention the "five-fold improvement in precision over measurements made in 2011." This is likely the most important take-away message in the entire paper, but is not even given in the abstract. (And, it should be noted, perhaps this number would have be 7.5x better, instead of 5, if not for the "noisy reference voltage"?) This statistic packs a punch, and would draw readers in.*

These performance comparisons were added in the abstract, summary and in the campaign discussions. Also I re-examined the comparison of the precisions of 2011 vs 2016 campaigns listed in the tables of the respective papers. These showed found the improvement in precision (for 1 second averaging) for 2016 vs 2011 was actually x9. The numbers in the text were updated correspondingly.

*2.4a Remarking on the feasibility of the CO2 Sounder lidar as a future space-based instrument in the abstract, as well, rather than just saying that this study serves as a demonstration of its feasibility, would be a good lead-in for potential readers.*

Updates were made to the abstract, discussion and summary.

*2.4b How close are you, in some kind of phase space, to technical readiness for a space-based version? Are you there on detectors, or are you a factor of 10 away? Are you there on laser power, or are you a factor of ten away?*

These aspects of the discussion section were updated as recommended.

*2.5 Throughout the rest of the text, I might recommend pulling numbers from earlier flights' similar-altitude flight legs to compare to those given here for the 2014 and 2016 flights. In the specific comments I list a few places where this can be done – including the summary.*

Comparisons were added.

*2.6 One other area which might benefit from further detail is the 2016 flight that used 15 wavelengths instead of 30 – the authors could explain that a space-based instrument will likely sample fewer wavelengths, so it is important to test the capabilities of the instrument with that change implemented. This would tie this data, once again, to the larger science goal. Speculation on why that sampling change may or may not make a difference in the results, and whether a change was expected, could also be given. The results of other flights using fewer than 30 wavelengths could also be mentioned – did they show any significant difference? It says later that a space-based instrument would only use 8. Why then are you testing with 15 or 30, and not 8?*

Although our early plans for using this technique in space used 8 wavelengths, the number of wavelengths we presently plan to use for space is 16. A sentence on this was added to the discussion section. This is why a similar number (15, which was much easier to change the existing airborne lidar to) was used for the 2016 flights. The discussion of the 2016 flight was also augmented with the rationale for using 15 wavelengths.

*2.7 Finally, more discussion of your ability to determine your biases is needed. A low bias is perhaps the most important potential advantage of the lidar over passive measurements. Can you get your biases down to 0.3 ppm? 0.1 ppm? Therefore, doing a good job to quantify the level at which you can determine your biases is a critical aspect to these test flights. It can make or*

*break a decision to do ASCENDS or not. From this paper, it appears that you cannot see any statistically significant biases in your retrieval (as compared to the in-situ). But what is the current level at which you can determine your biases? I note that while you are often within the standard deviation of the different lidar column measurements at a given altitude, you are virtually never within 1 standard deviation of the mean. If your errors are truly random, the errors should integrate down like the square root of N. Since you often have ~100 measurements per altitude bin, your noise-driven error is on the order 0.05-0.1 ppm on the mean XCO2 in an altitude bin, while your difference to the in-situ is often more like 0.5 ppm. That is a 5-to-10 sigma discrepancy (ie, it looks like a bias!). However, it could be that the in-situ columns are only good to 0.5 ppm, considering the change in the column that determines over the course of the spiral, the native in-situ errors, etc. Therefore, more commentary on and quantification of your biases and bias knowledge is sorely needed.*

The reviewer makes some very good points and we agree this is an important topic. A discussion on the differences between the XCO2 calculated from in situ and those from the lidar retrievals was added. This addresses some potential causes of differences and our plans to investigate them further.

*Specific comments:*
*2.8 Abstract: As stated above, please mention the factor of five improvement in precision in the 2016 flights compared to the 2011 version of the instrument.*

This was done and the improvement factor was recalculated and was updated.

*2.9 Section 1. Intro - When describing currently operational (passive) space-based instruments and their role in carbon cycle science, it might be informative to comment on their abilities to meet the sub-ppm measurement requirement. It would further tie the statistical results of this study to the larger science goal, which is in part to improve upon passive datasets.*

A few sentences were added on GOSAT and OCO-2 passive spectrometer missions and recent assessments of their performance.

*2.10 Section 2. Instrument background - Paragraph 3 of this section is the only place where the writing feels a bit disjointed and repetitive, and could be reorganized slightly to convey its message more clearly. It hops back and forth from spectroscopic information to environmental variables and Doppler-shift information a few times; I think that a simple rearrangement would help.*

Thank you – this text was rearranged.

*2.11 Section 3. 2014/16 Lidar setup - it is not clear in the text why the receiver optical transmission (in Table 1, which is referenced on page 3, line 45) was so low in 2014 (9.2%, as compared to ~50% and 60% in 2011, 2016). The authors should comment on this in this section. What specifically was done to improve it so much from 2014 to 2016 ?*

Comments were added addressing the cause of the low transmission in 2014.

*2.12 Section 4. Data processing & retrieval – The retrieval summary mentions that "HDO absorption spectrum can bias the retrieved XCO2 value if not taken into account," but it does not give the average value of that bias. The authors should include either that number or a reference to further information on the topic. Also, there seems to be no reference provided for a more detailed description of the retrieval process.*

*2.13 Finally, what is the posterior uncertainty on your measured HDO column? This could be a side benefit of your technique – knowledge of HDO can be used to study water cycle processes (see e.g. Frankenberg et al, 2013, "Water vapor isotopologue retrievals from high-resolution GOSAT shortwave infrared spectra", AMT).*

Two paragraphs were added about the HDO lines, their retrievals, and their potential use for other purposes.

*2.14 Also, do you fit for a scale factor to a prior CO2 profile, or do you assume a uniform CO2 profile and solve for that single concentration?*

This was clarified in the section on retrievals.

*2.15 Finally, and perhaps most importantly, is the short paragraph on the "clumped fitting" approach. This appears to be a novel aspect to this work. Has this approach of yours been described elsewhere in the literature (if so, please give the reference) or is it introduced here? How much does it reduce the errors (and in what way? Ie – is it simply a lowered scatter at a given altitude level?) between L2a and L2b? And how physically does it reduce the errors – what are the mechanisms? This could be an important aspect to this work, but it is downplayed – not mentioned at all in either the abstract or summary. Is is something that none of the passive teams (or other active teams, to my knowledge) currently do – you figured it out, so please make some more noise about it if it is really useful!*

Two paragraphs were added with more explanation about clumped fitting.

*2.16 Section 5. Campaigns overview – There are a few opportunities here to reference the precision and accuracy of previous campaign data. For example, on page 6, line 35, the authors could state that this <1ppm agreement between lidar and in situ is improved from 2011 comparisons (Abshire et al. 2013b), when this agreement was more along the lines of <1.4ppm.*

These were done.

*2.17 Also, more should be stated about the "noisy reference voltage". Does this imply that if this were fixed, the precision would be another 33% better? That's significant.*

After discussions among our team, we softened this statement to refer to noisy detector electronics. We also did change the electronics before our 2017airborne campaign to eliminate this potential source of noise. We plan to evaluate the change & re-compare to the model as we process our 2017 measurements.

*2.18 Section 6. I might contest, on the return leg (bottom of Figure 18), the statement "There is good agreement between E-W gradients measured by lidar and those computed from model", especially in the 5.6km segment. The model predicts something like a 1ppm E-W gradient, whereas the observational gradient might be closer to ~4ppm. Please soften this statement.*

Agree and the statement was rephrased and softened.

*2.19 SECTION 8. Discussion - Line 2: "Changing from 30 to 15 lines did not significantly change the retrievals, per the 2016 snow flight." It wouldn't hurt to include the numbers on this – how much of a difference did it make? Or was it truly negligible?*

A sentence was added that specifies the small differences between 30 and 15 wavelengths on this flight.

*2.20 SECTION 9. Summary – to restate what I said above, more quantification of your biases, and at what level you can even determine them with comparisons to in-situ, is needed. On line 28 would be meaningful to state specifically how much smaller the lidar vs. in situ biases are than in 2011, and how this was (missing word – assume "achieved" was intended).*

The updates requested above were made to the summary, as well as including more of the previous recommendations.

*2.21 Typos/grammar:*

Thank you again for such a careful reading. All the issues the reviewer noted below were corrected.

*a. Page 2, Line 4 – Cleaner wording – remove "so can" from "[…], and so can cause large retrieval errors."*

*b. Page 2, Line 22 – Define IPDA acronym once before using it in the rest of the text.*

*c. Page 2, Line 42 – Missing a period after "et al" in "[…] Abshire et al, 2013b)."*

*d. Page 5, Line 25 – Typo – "from based on" – remove either "from" or "based on".*

*e. Figure 7 – This figure appears blurry; update resolution.*

*f. Figure 9 – Change RMS labels to include "initial" and "after fit" for better at-a-glance clarity. Also would recommend, in the caption, describing figures in the order in which they appear - top then bottom.*

*g. Figure 14 & similar figures – If range and ground elevation are on the same axis (righthand), perhaps clarify this by making them the same color but making one dotted/dashed.*

*h. Figure 18 - The bottom plot doesn't show any 10.8km data, even though it's included in the legend. Is this a mistake? If so, correct plot; if not, removed 10.8km data from legend.*

*i. A secondary legend for observations vs. model would be better than using arrows.*
*j. The statistics in the legends don't seem to be described anywhere; describe in caption. Is the correlation between observations and model?*

*k. Page 8, Line 7 – Reference in wrong format? ("[54]")*

[revised manuscript text omitted]

---

## Author Response (AR2)

February 14, 2018

Dear Editors,

I made all the corrections that were recommended by the Guest Editor, as copied below.
I greatly appreciate the careful reviews and the many constructive recommendations – they have substantially improved the manuscript.

Sincerely,
Jim Abshire

Associate Editor Decision: Publish subject to technical corrections (13 Feb 2018) by William R. Simpson
Comments to the Author:
Thank you for addressing the points brought up by the reviewers. The manuscript nicely describes the current state of the instrument and improvements from the 2011 version. I saw two small points in my final read, which are listed below:

Page 2, line 26: "to better define" is a split infinitive. It should say "to define the science and measurement needs better and..."

Table 1: A few cells have a green triangle at the top left. Please remove these. Consider changing "Detector APD gain settings" to "Detector gain" because 2011 didn't use an APD, but a PMT instead.